# Tackling Length Inflation Without Trade-offs: Group Relative Reward Rescaling for Reinforcement Learning

**Zichao Li** [1 2]  **Jie Lou** [3]  **Fangchen Dong** [3]  **Zhiyuan Fan** [3]  **Mengjie Ren** [1 2]  **Hongyu Lin** [1]  **Xianpei Han** [1]
**Debing Zhang** [3]  **Le Sun** [1]  **Yaojie Lu** [1]  **Xing Yu** [3]

## Abstract

Reinforcement learning significantly enhances LLM capabilities but suffers from a critical issue: *length inflation*, where models adopt verbosity or inefficient reasoning to maximize rewards. Prior approaches struggle to address this challenge in a general and lossless manner, primarily because additive penalties introduce a compensatory effect that creates optimization shortcuts, while heuristic gating strategies lack generality beyond binary feedback. To bridge this gap, we present **Group Relative Reward Rescaling (GR³)**, which reframes length control as a multiplicative rescaling paradigm, effectively establishing a generalized, continuous, and reward-dependent gating mechanism. To further ensure lossless optimization, we incorporate group-relative regularization and advantage-aware calibration, which dynamically adapt length budgets to instance difficulty and preserve the advantage signal of high-quality trajectories. Empirically, across both RLHF and RLVR settings, GR³ maintains training dynamics and downstream performance comparable to standard GRPO while significantly mitigating length inflation, outperforming state-of-the-art length-regularized baselines.

## 1. Introduction

Reinforcement learning (RL) (Bai et al., 2022; Guo et al., 2025) has become the engine of post-training for Large Language Models (LLMs) (Achiam et al., 2023; Team et al., 2023). Yet this engine exhibits a persistent flaw, which we

---
[1]Chinese Information Processing Laboratory, Institute of Software, Chinese Academy of Sciences [2]University of Chinese Academy of Sciences [3]Xiaohongshu Inc. Correspondence to: Jie Lou <loujie0822@gmail.com>, Yaojie Lu <luyaojie@iscas.ac.cn>.

*Proceedings of the 43rd International Conference on Machine Learning*, Seoul, South Korea. PMLR 306, 2026. Copyright 2026 by the author(s).

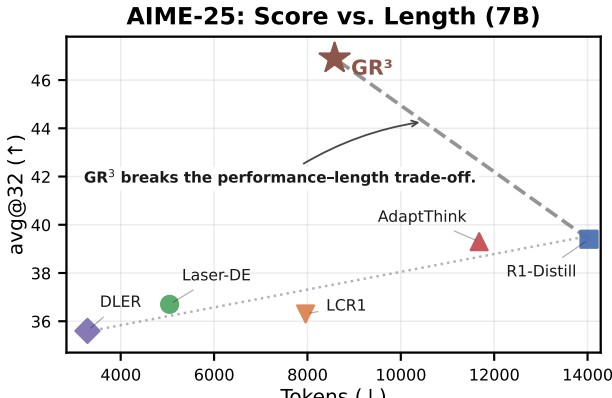

*Figure 1.* Comparison of GR³ with open-source efficient reasoning models, all trained on DeepSeek-R1-Distill-7B. **GR³ sustains stable performance gains under RL while simultaneously mitigating the length inflation issue.**

term *length inflation*: a tendency for RL-trained models to produce unnecessarily lengthy trajectories, inflating inference costs without proportional gains in quality. This phenomenon arises across major RL paradigms. In RL from human feedback (RLHF) (Ouyang et al., 2022), models exploit reward-model biases toward verbosity, leading to reward hacking (Gao et al., 2023). In RL with verifiable rewards (RLVR) (Shao et al., 2024), length inflation instead stems from reasoning inefficiency (Sui et al., 2025), where models generate unnecessarily long chains of thought to marginally improve the likelihood of a correct solution.

Prior work has sought to mitigate length inflation. One line of research trains reward models that are invariant to response length (Chen et al., 2024a; Liu et al., 2024). While effective in RLHF, this strategy does not extend to RLVR, where rewards are derived from ground-truth verifiers rather than learned proxies that can be debiased. A more general direction instead introduces explicit length penalties into the reward (Luo et al., 2025a; Liu et al., 2025c; Yi et al., 2025). However, most existing methods rely on coarse regularization, which leads to suboptimal optimization dynamics.

A common design adopts additive shaping (Yu et al., 2025; Team et al., 2025), modifying the objective with an explicit length term (e.g., $R' = R - \lambda\ell$). This introduces decou-

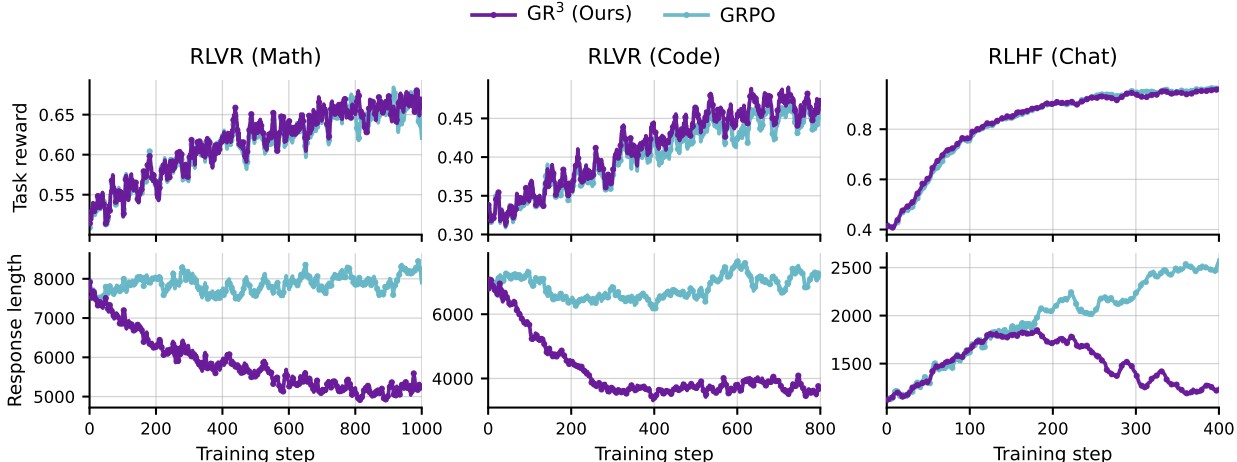

*Figure 2.* Training dynamics of GR$^3$, which retains GRPO's reward gains without loss while significantly reducing average tokens. The base models used for the two settings (i.e., RLVR and RLHF) are DeepSeek-R1-Distill-1.5B and Qwen3-8B, respectively.

pled incentives, creating a length-driven component that makes extreme brevity an attractive shortcut independent of task success. To better align penalties with outcomes, some works propose heuristic gating (Cheng et al., 2025; Arora & Zanette, 2025), applying penalties only when $R = 1$. However, such designs are inherently limited to binary feedback and do not extend naturally to continuous-reward settings like RLHF. Moreover, many approaches rely on coarse control mechanisms, such as static truncation thresholds or uncalibrated penalty strengths (Liu et al., 2025b; Cheng et al., 2025), resulting in an inherent efficiency–performance trade-off, as illustrated in Figure 1.

These observations lead to a central question: *Can we tackle length inflation in a general manner without compromising the capability gains of RL?* In this work, we present **Group Relative Reward Rescaling (GR$^3$)**, a principled framework for lossless efficiency optimization. Rather than using additive penalties, GR$^3$ regularizes length through **multiplicative rescaling**, which acts as a generalized gating mechanism and removes the compensatory shortcuts inherent to additive schemes. To further ensure lossless optimization, we introduce two fine-grained mechanisms. Specifically, we employ **group-relative regularization**, which normalizes length against on-policy statistics rather than rigid thresholds, thereby dynamically adapting the length budget to the inherent difficulty of each prompt. Complementing this, we introduce **advantage-aware calibration** to explicitly control the penalty strength. This ensures that length regularization does not overturn the advantage signal of representative high-quality trajectories, thereby safeguarding stable optimization toward capability improvements.

Empirically, GR$^3$ resolves the efficiency–performance trade-off inherent in prior methods. As shown in Figure 1, our

approach significantly reduces token usage (e.g., over $40\%$ on AIME-25) while simultaneously improving accuracy (e.g., $+8$ points), demonstrating that verbosity is not a prerequisite for intelligence. Furthermore, in RLHF settings, GR$^3$ exhibits an adaptive length dynamic: it permits moderate growth when computation is beneficial but automatically curtails generation length as the policy matures (Figure 2). This mechanism effectively mitigates reward hacking via verbosity without sacrificing capability.

In summary, our contributions are threefold:

- We propose GR$^3$, a framework for lossless length control that substitutes additive penalties with multiplicative reward rescaling. This design eliminates compensatory optimization shortcuts and provides a unified mechanism for both binary and continuous rewards.

- We develop an optimization-preserving strategy that integrates group-relative regularization with advantage-aware calibration, adapting constraints to on-policy statistics while preserving learning signals.

- Across mathematical reasoning, code generation, and RLHF alignment tasks, GR$^3$ yields concise generations while matching standard GRPO performance, shifting the efficiency–performance Pareto frontier.

## 2. Preliminary

### 2.1. Group Relative Policy Optimization

LLM generation can be formulated as a token-level Markov Decision Process (Puterman, 1990). Given a prompt $x \sim \mathcal{D}$, an autoregressive policy $\pi_\theta$ generates a response $y = (y_1, \ldots, y_\ell)$ of length $\ell := |y|$ by sampling tokens from

$\pi_\theta(y_t \mid x, y_{<t})$. A scalar reward $R(x, y)$ is defined over complete responses, and reinforcement learning aims to maximize the expected reward:

$$\max_{\pi_\theta} \mathbb{E}_{x \sim \mathcal{D}, y \sim \pi_\theta(\cdot|x)}\big[R(x, y)\big]. \quad (1)$$

With the emergence of reasoning models such as DeepSeek-R1 (Guo et al., 2025), group-style RL has become prevalent for LLM post-training. Among them, Group Relative Policy Optimization (GRPO) (Shao et al., 2024) is widely adopted due to its scalability and its elimination of the need for a separate value model. For each prompt $x$, GRPO samples a group of $G$ responses $\{y^{(i)}\}_{i=1}^G$ from an old policy $\pi_{\theta_{old}}(\cdot \mid x)$ and evaluates each by $R(x, y^{(i)})$. It then constructs a group-relative advantage via within-group normalization:

$$\hat{A}^{(i)} = \frac{R(x, y^{(i)}) - \mu_R}{\sigma_R}$$

$$\mu_R := \frac{1}{G}\sum_{j=1}^G R(x, y^{(j)}), \sigma_R := \text{std}\Big(\{R(x, y^{(j)})\}_{j=1}^G\Big) \quad (2)$$

The policy is optimized using a PPO-style clipped objective over the group-normalized advantages:

$$\mathcal{J}_{\text{GRPO}}(\theta) = \mathbb{E}_{x \sim \mathcal{D}, \{y^{(i)}\}_{i=1}^G}\Bigg[\frac{1}{G}\sum_{i=1}^G \sum_{t=1}^{|y^{(i)}|}$$

$$\Big(\min\big(r_{i,t}(\theta)\hat{A}^{(i)}, \text{clip}(r_{i,t}(\theta), 1-\varepsilon, 1+\varepsilon)\hat{A}^{(i)}\big) \quad (3)$$

$$- \beta D_{\text{KL}}(\pi_\theta \| \pi_{\text{ref}})\Big)\Bigg],$$

where the importance sampling ratio is defined as

$$r_{i,t}(\theta) = \frac{\pi_\theta(y_t^{(i)} \mid x, y_{<t}^{(i)})}{\pi_{\theta_{old}}(y_t^{(i)} \mid x, y_{<t}^{(i)})}. \quad (4)$$

Notably, GRPO estimates advantages based on group-wise statistics. We exploit this structural property to construct a more on-policy length regularization scheme and to formulate advantage-aware calibration that better respect the underlying optimization signal, as detailed in Section 3.

## 2.2. Length-Regularized Reinforcement Learning

Despite the substantial performance gains brought by reinforcement learning (RL), a critical failure mode has become increasingly apparent, which we refer to as *length inflation*. Long-CoT models are especially susceptible to overthinking (Chen et al., 2024b; Luo et al., 2025a). In parallel, reward hacking (Gao et al., 2023; Eisenstein et al., 2023) in RLHF can also lead to explosive growth in response length.

*Table 1.* Comparison of length-aware reward shaping methods (Liu et al., 2025c; Arora & Zanette, 2025; Aggarwal & Welleck, 2025; Team et al., 2025; Yu et al., 2025; Cheng et al., 2025). $\ell^{(i)}$ is the response length of sample $i$. $\ell_{\min}$, $\ell_{\max}$, $\bar{\ell}$, and $\sigma_\ell$ are computed within each group. $\alpha$, $\lambda$, $\ell_T$ and $\ell_C$ are fixed hyperparameters. $s(\cdot)$ denotes the sigmoid function.

| Method | $S$ (Length shaping term) |
|---|---|
| **Additive:** | $\hat{R}^{(+)} = R + \lambda \cdot S$ |
| L1-Exact | $-\|\ell^{(i)} - \ell_T\|$ |
| DAPO | $\begin{cases} 0, & \ell^{(i)} \le \ell_T - \ell_C, \\ \frac{\ell_T - \ell_C - \ell^{(i)}}{\ell_C}, & \ell_T - \ell_C < \ell^{(i)} \le \ell_T, \\ -1, & \ell^{(i)} > \ell_T \end{cases}$ |
| Kimi-k1.5 | $\begin{cases} 0.5 - \frac{\ell^{(i)} - \ell_{\min}}{\ell_{\max} - \ell_{\min}}, & R = 1, \\ \min\left(0.5 - \frac{\ell^{(i)} - \ell_{\min}}{\ell_{\max} - \ell_{\min}}, 0\right), & R = 0 \end{cases}$ |
| Truncation | $-\mathbb{I}(R=1) \cdot \mathbb{I}(\ell^{(i)} > \ell_T)$ |
| Efficiently | $-\mathbb{I}(R=1) \cdot s\left((\ell^{(i)} - \bar{\ell})/\sigma_\ell\right)$ |
| LC-R1 | $\mathbb{I}(R=1) \cdot (1 - \ell^{(i)}/\ell_{\max})$ |
| **Multiplicative:** | $\hat{R}^{(\times)} = R \cdot S$ |
| GR$^3$ *(Ours)* | $\frac{1}{1 + \alpha \cdot \frac{\ell^{(i)}}{\bar{\ell}}}$ |

A common strategy for mitigating length inflation in reinforcement learning is to explicitly regularize response length through reward shaping. From a unified perspective (Liu et al., 2025c), Most existing approaches can be instantiated as *additive shaping*:

$$\textbf{Additive:} \quad \hat{R}^{(+)} = R + \lambda \cdot S, \ \lambda > 0 \quad (5)$$

where $R$ is the task reward, $S$ is a length-dependent shaping signal, and $\lambda$ controls the strength of length regularization. Here, $\hat{R}^{(+)}$ denotes the shaped reward used for efficient policy optimization. We present several representative examples for illustration, as shown in Table 1.

Under this formulation, existing approaches mainly differ in how the length regularizer $S$ is instantiated. The most basic strategy typically relies on a fixed threshold (e.g., $\ell_T = 4K$): once the response length exceeds this limit, the model incurs either a constant penalty (Liu et al., 2025b) or a progressively increasing one (Aggarwal & Welleck, 2025). A more principled paradigm instead leverages group-level statistics (Team et al., 2025) to determine the strength of the penalty. In addition, some methods introduce gating mechanisms (e.g., $\mathbb{I}(R=1)$) (Arora & Zanette, 2025), which activate length regularization only for successful trajectories, so as to prevent the model from over-optimizing for brevity.

**Multiplicative vs. Additive Shaping**

*Figure 3.* Additive length regularization consistently degrades task reward across different choices of $\lambda$, whereas the multiplicative scheme maintains stable reward improvement throughout training.

While these methods vary in their specific instantiations, they share a common objective: compressing response length during RL training. Empirically, however, such regularization often leads to performance degradation, motivating a closer examination of reward shaping design.

## 3. Group Relative Reward Rescaling

In this section, we introduce **Group Relative Reward Rescaling (GR$^3$)**, a principled framework designed to mitigate length inflation without compromising capabilities.

Formally, for a response $y^{(i)}$ with length $\ell^{(i)} = |y^{(i)}|$ within a group of $G$ samples, GR$^3$ defines the rescaled reward as:

$$\hat{R}(x, y^{(i)}) = R(x, y^{(i)}) \cdot \underbrace{\frac{1}{1 + \alpha \cdot \frac{\ell^{(i)}}{\bar{\ell}}}}_{S^{(i)}} \qquad (6)$$

Here, $S^{(i)}$ represents the length scaling factor, and $\bar{\ell}$ denotes the average response length within the group.

This formulation reflects a unified design along three dimensions. We adopt **multiplicative reward rescaling** (Section 3.1) as a reward-dependent gate, mitigating the compensation effect while remaining applicable to general continuous reward distributions. We instantiate the scaling factor via **group-relative regularization** (Section 3.2), which utilizes the on-policy average $\bar{\ell}$ to dynamically adapt the length budget to the inherent difficulty of the prompt. To maintain stable optimization, **advantage-aware calibration** (Section 3.3) further controls penalty strength, preventing the undue suppression of high-advantage trajectories.

### 3.1. Multiplicative Reward Rescaling

#### 3.1.1. MOTIVATION AND FORMULATION

The most naive length control methods rely on additive reward shaping as Eq. 5, where $\lambda$ controls the strength of the length penalty. However, **additive shaping introduces an**

**inherent compensatory effect**: the shaping term $S$ forms an auxiliary optimization objective that can be exploited independently of task performance. Several works mitigate this issue via gating mechanisms, e.g., incorporating $I(R = 1)$ into $S$. Such designs, however, are restricted to binary rewards and do not extend to continuous reward settings.

Departing from these additive formulations, we propose a more general *multiplicative shaping* paradigm:

$$\textbf{Multiplicative:} \quad \hat{R}^{(\times)} = R \cdot S, \qquad (7)$$

which can be interpreted as a continuous extension of heuristic gating, eliminating the trade-off coefficient $\lambda$ and naturally generalizing to arbitrary reward scales [1]. Crucially, the multiplicative formulation removes the compensatory property of additive shaping, requiring the policy to jointly optimize task performance and length control. As shown in Figure 3, additive shaping exhibits a systematic failure mode: for any choice of $\lambda$, the optimization is dominated by rapid length reduction, resulting in severe performance degradation. Multiplicative shaping, by contrast, does not admit such a shortcut and therefore avoids this collapse.

Intuitively, multiplicative shaping couples the influence of length control to the task reward through

$$\frac{\partial \hat{R}^{(\times)}}{\partial S} = R. \qquad (8)$$

Thus, length regularization automatically strengthens with task success, making the shaping intrinsically reward-aware.

#### 3.1.2. ANALYSIS UNDER GROUP-NORMALIZED ADVANTAGE

We further refine the above intuition under the group-normalized advantage adopted in GRPO (Eq. 2). Fix a prompt $x$, and consider the within-group distribution over random variables $(R, S)$ induced by sampling responses $y \sim \pi_{\theta_{\text{old}}}(\cdot \mid x)$, where $R \in [0, 1]$ denotes the task reward and $S \in [0, 1]$ denotes a length-related score (e.g., Eq. 6).

We compare additive shaping $\hat{R}^{(+)}$ (Eq. 5) and multiplicative shaping $\hat{R}^{(\times)}$ (Eq. 7). Let $\mu.$ and $\sigma.$ denote within-group mean and standard deviation. Results are stated using population moments; empirical versions follow by substituting sample averages.

**Proposition 3.1** (Additive shaping: linear injection of the length signal)**.** *Let* $(R, S)$ *have finite second moments and define* $\mu_R = \mathbb{E}[R]$, $\mu_S = \mathbb{E}[S]$, $\sigma_R^2 = \text{Var}(R)$, $\sigma_S^2 = \text{Var}(S)$, *and* $\sigma_{RS} = \text{Cov}(R, S)$. *For* $\hat{R}^{(+)} = R + \lambda S$ *with* $\lambda > 0$,

$$\hat{R}^{(+)} - \mathbb{E}[\hat{R}^{(+)}] = (R - \mu_R) + \lambda(S - \mu_S), \qquad (9)$$

---

[1]In Appendix A, we discuss the connection between heuristic gating and multiplicative rescaling, and show that heuristic gating fails in continuous-reward RLHF settings.

$$\text{Var}(\hat{R}^{(+)}) = \sigma_R^2 + \lambda^2 \sigma_S^2 + 2\lambda \sigma_{RS}, \qquad (10)$$

*and hence*

$$A\left(\hat{R}^{(+)}\right) = \frac{(R - \mu_R) + \lambda(S - \mu_S)}{\sqrt{\sigma_R^2 + \lambda^2 \sigma_S^2 + 2\lambda \sigma_{RS}}}. \qquad (11)$$

*Therefore, the length-related signal $(S - \mu_S)$ is linearly injected into the advantage with fixed weight $\lambda$, and can contribute even when $R$ provides little discriminative signal within the group.*

*Proof.* $\mathbb{E}[\hat{R}^{(+)}] = \mu_R + \lambda \mu_S$ gives Eq. (9). Eq. (10) follows from $\text{Var}(X+Y) = \text{Var}(X) + \text{Var}(Y) + 2\text{Cov}(X,Y)$ with $X = R$ and $Y = \lambda S$. Eq. (11) follows by applying Eq. (2) with $R$ replaced by $\hat{R}^{(+)}$. $\qquad\square$

**Proposition 3.2** (Multiplicative shaping: reward-weighted length signal). *Let $(R, S)$ have finite second moments and define $\mu_R = \mathbb{E}[R]$, $\mu_S = \mathbb{E}[S]$, $\sigma_R^2 = \text{Var}(R)$, $\sigma_S^2 = \text{Var}(S)$, and $\sigma_{RS} = \text{Cov}(R, S)$. For multiplicative shaping $\hat{R}^{(\times)} = RS$,*

$$\mathbb{E}[\hat{R}^{(\times)}] = \mathbb{E}[RS] = \mu_R \mu_S + \sigma_{RS}. \qquad (12)$$

*Moreover, the centered shaped reward admits the decomposition*

$$RS - \mathbb{E}[RS] = R(S - \mu_S) + \mu_S(R - \mu_R) - \sigma_{RS}. \qquad (13)$$

*Consequently, the group-normalized advantage can be written as*

$$A\left(\hat{R}^{(\times)}\right) = \frac{R(S - \mu_S) + \mu_S(R - \mu_R) - \sigma_{RS}}{\sqrt{\text{Var}(RS)}}. \qquad (14)$$

*Proof.* Eq. (12) follows from $\mathbb{E}[RS] = \mathbb{E}[R]\mathbb{E}[S] + \text{Cov}(R, S)$. For Eq. (13), rewrite

$$RS = R\mu_S + R(S - \mu_S) = \mu_R \mu_S + \mu_S(R - \mu_R) + R(S - \mu_S),$$

and subtract $\mathbb{E}[RS] = \mu_R \mu_S + \sigma_{RS}$. Eq. (14) is Eq. (2) applied to $\hat{R}^{(\times)}$. $\qquad\square$

*Remark* 3.3 (Why multiplicative shaping is reward-aware under group normalization). Under additive shaping, Proposition 3.1 shows that the length deviation $(S - \mu_S)$ is injected into the centered shaped reward with a fixed coefficient $\lambda$. This creates a compensatory degree of freedom: the policy can improve the shaped reward by manipulating $S$ even when $R$ provides little learning signal.

In contrast, Proposition 3.2 yields the decomposition

$$RS - \mathbb{E}[RS] = R(S - \mu_S) + \mu_S(R - \mu_R) - \sigma_{RS},$$

where the influence of length deviation is scaled by $R$ itself. Consequently, length control is weak when rewards are low and strengthens as task performance improves, making multiplicative shaping inherently reward-aware.

*Table 2.* Comparison of standard RLVR training, threshold-based truncation (Hou et al., 2025), and group-relative methods (Arora & Zanette, 2025). All methods are evaluated at the 800th step.

| Model | AIME24 | | MATH500 | |
|---|---|---|---|---|
| | Score↑ | #Token↓ | Score↑ | #Token↓ |
| *Standard* | | | | |
| GRPO | 41.0 | 12885 | 87.2 | 5788 |
| *Threshold-based* | | | | |
| Truncation–4k | 32.5 | 3419 | 86.5 | 1438 |
| Truncation–8k | 34.8 | 6684 | 87.6 | 2918 |
| *Group-relative* | | | | |
| Efficiently-0.01 | 41.4 | 9544 | 76.0 | 1349 |
| Efficiently-0.2 | 41.3 | 10412 | 60.5 | 1265 |
| GR$^3$ *(ours)* | 44.0 | 8760 | 88.7 | 2405 |

### 3.2. Group Relative Length Regularization

Many prior works address length inflation by imposing absolute length thresholds (Yu et al., 2025; Liu et al., 2025c;b), penalizing trajectories that exceed a fixed budget. However, such designs may suppress necessary reasoning on hard instances, making the policy insensitive to task difficulty and degrading performance. More importantly, **a fixed threshold inevitably induces off-policy bias**: the optimal reasoning length varies across tasks and shifts during training, which a single global constant cannot capture.

To overcome these limitations, we propose a group-relative length regularization strategy that adapts to on-policy behavior. Following Eq. 6, we define a bounded length-shaping term $S^{(i)} \in (0, 1)$ using within-group statistics:

$$S^{(i)} = \frac{1}{1 + \alpha \cdot \frac{\ell^{(i)}}{\bar{\ell}}}, \qquad \alpha > 0. \qquad (15)$$

where $\ell^{(i)}$ is the response length and $\bar{\ell}$ is the group mean. This penalty decreases smoothly with length, while normalizing against $\bar{\ell}$ avoids arbitrary global thresholds and adapts to the model's current generation behavior.

As shown in Table 2, we include the fixed-threshold truncation method (Hou et al., 2025) as a minimalist baseline. We find that threshold-based truncation imposes a uniform maximum response length even on difficult benchmarks, which degrades reasoning performance on challenging problems. We also compare against other group-relative methods and find that certain shaping strategies (Arora & Zanette, 2025) introduce biases that favor shallow reasoning on easier benchmarks (see Appendix B for analysis). We additionally evaluated another group-relative method, Kimi-k1.5 (Team et al., 2025), but it exhibited training collapse; therefore, we omit its results. We attribute this failure to the additive shaping paradigm without gating, as discussed in Section 3.1.

### 3.3. Advantage-Aware Calibration

Within the framework of group-relative policy optimization, the length penalty term $S$ acts as a powerful shaper of the advantage landscape. The interaction between the penalty strength and group normalization is non-trivial: **slight shifts in $S$ can noticeably redirect the optimization trajectory**. In practice, an unconstrained or overly strong penalty may penalize high-quality responses so heavily that it creates a contradictory signal where the model is discouraged from generating its best responses.

A natural but overly strict objective is to require that *all* high-quality trajectories retain positive advantages. However, this becomes intractable under *high reward density*, where most responses in a group achieve the maximum reward $R_{\max}$ (e.g., 15 out of 16 are correct). Due to the zero-sum structure of group normalization, correct but above-average-length responses may inevitably receive negative advantages. We provide a formal analysis of this limitation in Appendix C.1.

**Average-Case Advantage Preservation.** Instead of protecting the longest outlier trajectory, we aim to preserve the advantage of a representative high-quality response. Specifically, we consider a hypothetical response that achieves the group-wise maximum reward $R_{\max}$ with the group-average length $\bar{\ell}$, and require its advantage to remain non-negative. Let $\mu_{\hat{R}}$ denote the mean regularized reward in the group. This yields the condition:

$$\frac{R_{\max}}{1 + \alpha \cdot \frac{\bar{\ell}}{\bar{\ell}}} \geq \mu_{\hat{R}} \quad \implies \quad \frac{R_{\max}}{1 + \alpha} \geq \mu_{\hat{R}} \qquad (16)$$

This ensures that the penalty $\alpha$ does not overturn the advantage of a typical high-quality response. In the limiting case where all trajectories in a group achieve $R_{\max}$, the average-case constraint can still fail to hold. We therefore filter out such groups online (see Appendix C.2).

In practice, Eq. (16) is not enforced as a hard constraint at every update due to on-policy sampling stochasticity. Instead, we treat it as a calibration criterion for selecting the penalty coefficient $\alpha$. We run a short calibration phase at the beginning of GRPO training and measure the *Constraint Satisfaction Rate* (CSR) over candidate $\alpha$ values. We then select the largest $\alpha$ whose CSR remains consistently high (e.g., $\geq 99.9\%$), ensuring high-probability constraint satisfaction while maintaining strong length regularization.

Empirically, the $\alpha$ selected by this protocol maintains a near-perfect CSR throughout training (see Figure 4). This point effectively marks a practical boundary: the reward gap relative to the GRPO baseline is already positive, indicating preserved task capability. Further reducing the penalty strength (i.e., decreasing $\alpha$) yields no consistent performance gains, and instead results in fluctuations consistent with training variance.

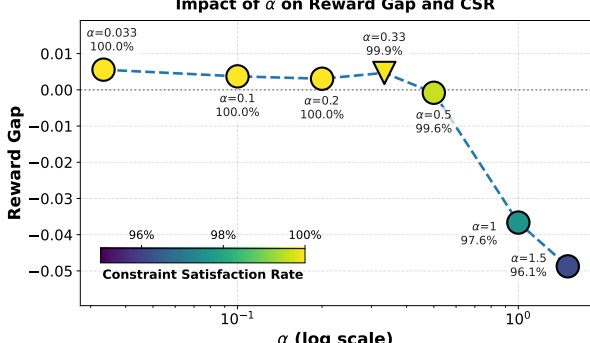

*Figure 4.* Sensitivity of $\alpha$: reward gap relative to the standard GRPO baseline versus $\alpha$ (log scale). Marker color indicates the average CSR measured during actual training, while the triangle marker denotes the value of $\alpha$ selected during the calibration phase.

## 4. Experiments

### 4.1. Setup

**Efficient Reasoning for RLVR.** Following prior work, we adopt DeepSeek-R1-Distill-1.5B and DeepSeek-R1-Distill-7B (Guo et al., 2025) as the base models. For mathematical reasoning, we use the DeepScaleR-Preview-Dataset (Luo et al., 2025b) as training data, and include open-sourced checkpoints of existing efficient-reasoning methods as baselines, including LC-R1 (Cheng et al., 2025), Laser (Liu et al., 2025c), AdaptThink (Zhang et al., 2025), and DLER (Liu et al., 2025b). To demonstrate the generality of our approach, we further extend to code generation, using the prompts from DeepDistill (Tian et al., 2025).

**Mitigating Length Bias in RLHF.** As for the RLHF setting, we use the non-reasoning versions of Qwen3-4B and Qwen3-8B (Yang et al., 2025) as base models. We construct RL prompts from arena-human-preference-140k[2], and employ Skywork-Reward-V2-Llama-3.1-8B (Liu et al., 2025a) as the reward model. To improve training stability, we apply a reference-based sigmoid shaping (Fu et al., 2025) scheme:

$$R(x, y^{(i)}) = s\left(R_{\text{origin}}(x, y^{(i)}) - R_{\text{origin}}(x, y^{\text{ref}})\right). \quad (17)$$

where $R_{\text{origin}}(\cdot)$ denotes the raw reward model score.

Detailed experimental settings are provided in Appendix D.

### 4.2. Main Results

#### 4.2.1. EFFICIENT REASONING FOR RLVR

The experimental results for the 7B model are presented in Table 3, while those for the 1.5B model are provided in Appendix E.1. Notably, **GR$^3$ improves reasoning performance while reducing generation length, indicating a**

---

[2] https://huggingface.co/datasets/lmarena-ai/arena-human-preference-140k

*Table 3.* Mathematical reasoning performance for 7B models. Length-oriented methods reduce tokens but may sacrifice accuracy, while **GR$^3$ consistently achieves comparable accuracy with significantly fewer tokens, establishing a markedly better Pareto frontier.**

| | AIME24 | | AIME25 | | AMC23 | | MATH500 | |
|---|---|---|---|---|---|---|---|---|
| Model | Avg@32↑ | #Token↓ | Avg@32↑ | #Token↓ | Avg@16↑ | #Token↓ | Avg@4↑ | #Token↓ |
| *Initial model* | | | | | | | | |
| DeepSeek–R1–Distill–7B | 52.4 | 13213 | 39.4 | 14032 | 89.8 | 6385 | 92.1 | 3994 |
| *Length-oriented RL* | | | | | | | | |
| LCR1–7B | 47.9 | 7548 | 36.3 | 7960 | 85.8 | 2963 | 89.1 | 1546 |
| Laser–DE–L4096–7B | 51.7 | 4931 | 36.7 | 5048 | 88.1 | 2427 | 92.4 | 1580 |
| AdaptThink–7B | 51.5 | 11070 | 39.3 | 11678 | 88.1 | 4280 | 90.6 | 2011 |
| DLER–R1–7B | 49.5 | 3272 | 35.6 | 3288 | 91.4 | 2255 | 93.2 | 1650 |
| *Performance-oriented RL* | | | | | | | | |
| GRPO | 57.1 | 11079 | 44.7 | 12540 | 90.3 | 7256 | 92.1 | 5006 |
| GR$^3$ *(ours)* | 60.1 | 7923 | 46.9 | 8582 | 93.0 | 3090 | 94.0 | 1764 |

*Table 4.* Performance on code generation tasks. **GR$^3$ achieves competitive scores with fewer tokens across model sizes.**

| | LiveCodeBench v6 | | MultiPL-E | |
|---|---|---|---|---|
| Model | Score↑ | #Token↓ | Score↑ | #Token↓ |
| *DeepSeek–R1–Distill–1.5B* | | | | |
| Initial | 17.7 | 12665 | 45.1 | 6181 |
| GRPO | 23.4 | 11830 | 51.5 | 6589 |
| GR$^3$ *(ours)* | 24.9 | 8538 | 52.2 | 2414 |
| *DeepSeek–R1–Distill–7B* | | | | |
| Initial | 37.7 | 11496 | 69.7 | 4121 |
| GRPO | 42.4 | 10956 | 71.1 | 4794 |
| GR$^3$ *(ours)* | 41.6 | 7504 | 70.9 | 2127 |

*Table 5.* Performance on chat tasks. GRPO improves performance but incurs substantial length bias, while **GR$^3$ achieves stronger alignment gains while preserving the length of the initial policy**.

| | Arena–Hard–Auto | | Alpaca–Eval | |
|---|---|---|---|---|
| Model | Score↑ | #Token↓ | Score↑ | #Token↓ |
| *Qwen3–4B* | | | | |
| Initial | 66.6 | 1139 | 40.1 | 737 |
| GRPO | 85.8 | 2374 | 33.9 | 1993 |
| GR$^3$ *(ours)* | 85.9 | 1377 | 44.1 | 859 |
| *Qwen3–8B* | | | | |
| Initial | 77.2 | 1171 | 50.2 | 778 |
| GRPO | 90.6 | 2343 | 53.5 | 1670 |
| GR$^3$ *(ours)* | 92.8 | 1178 | 55.8 | 765 |

**genuine efficiency gain rather than a trade-off.**

Within the performance-oriented regime, compared with standard GRPO, GR$^3$ leads to substantially shorter generations while maintaining or even improving performance. For instance, at the 7B scale on AIME24, GR$^3$ reduces the average length from 13,213 to 7,923 tokens while improving Avg@32 from 52.4 to 60.1, whereas GRPO only reaches 57.1. In contrast to existing length-oriented baselines, GR$^3$ does not over-compress the reasoning length at the cost of accuracy; instead, it prioritizes preserving performance while removing redundant reasoning. For example, on AIME25 (7B), none of the length-oriented baselines surpasses the initial checkpoint performance (39.4), whereas GR$^3$ improves it to 46.9 with much fewer tokens (14,032 → 8,582). **This suggests that GR$^3$ encourages more efficient reasoning trajectories rather than merely truncating reasoning, yielding consistent gains across scales.**

We further validate GR$^3$ on code generation benchmarks, as summarized in Table 4. Consistent with our findings in mathematical reasoning, GR$^3$ achieves substantial effi-

ciency gains while preserving task performance.

### 4.2.2. MITIGATING LENGTH BIAS IN RLHF

Results on alignment benchmarks are shown in Table 5. Compared with the initial models, RLHF training yields substantial improvements in chat quality. However, we observe that standard GRPO suffers from severe reward hacking under length bias, where the model can artificially increase reward by generating unnecessarily long responses, resulting in explosive length inflation (e.g., on Qwen3-8B, the average response length on Arena–Hard–Auto increases from 1,171 to 2,343 tokens). In contrast, **GR$^3$ attains comparable or even stronger alignment gains while keeping response length almost unchanged, effectively decoupling performance improvement from verbosity**. For instance, on Qwen3-8B, GR$^3$ improves the Arena–Hard–Auto score from 77.2 to 92.8, while the token cost only increases marginally (1,171 → 1,178).

We further visualize the training dynamics in the RLHF setting in Figure 2. Under GRPO, response length grows

monotonically and uncontrollably throughout training. In contrast, $GR^3$ exhibits a clear "increase-then-decrease" pattern: the model initially expands its reasoning to secure alignment improvements, and subsequently compresses redundant generations once performance stabilizes. This dynamic behavior aligns with our design intuition—**$GR^3$ prioritizes achieving reliable alignment gains first, and then progressively improves response efficiency by suppressing length-based exploitation**.

## 4.3. Analysis and Discussion

### 4.3.1. ABLATION ON PENALTY STRENGTH $\alpha$

We study the effect of the penalty coefficient $\alpha$ by sweeping its value while keeping all other settings fixed. Detailed results and analysis are provided in Appendix E.2; here we summarize the key findings.

When $\alpha$ is too large (e.g., 1.0), $GR^3$ degenerates toward a naive length regularizer: responses become much shorter, but performance gains over the base model are limited, as optimization is dominated by compression rather than capability improvement. As $\alpha$ decreases, response length grows smoothly, while task performance first improves and then plateaus. This trend is consistent with the analysis in Section 3.3: **once the advantage of representative high-quality trajectories is preserved, further reducing the penalty mainly relaxes length control without yielding stronger learning signals.** The chosen value $\alpha = 0.33$ lies near this transition region, achieving substantial length reduction while retaining most performance gains.

### 4.3.2. ABLATION ON DESIGN COMPONENTS

We further conduct a controlled ablation to disentangle multiplicative shaping, group-relative regularization, and advantage-aware calibration. Detailed results are provided in Appendix E.3.

Removing multiplicative shaping and reverting to additive penalties leads to severe accuracy degradation (average accuracy drops from 59.6 to 48.0), confirming that the multiplicative formulation is essential for preserving the learning signal structure. Retaining multiplicative shaping but replacing our group-relative term with other alternatives yields moderate degradation, with less effective length adaptation across benchmarks of varying difficulty. Removing advantage-aware calibration while keeping the other two components exposes a strong sensitivity to $\alpha$: overly large values tend to over-compress responses and hurt accuracy, whereas overly small values preserve accuracy but yield only marginal length reduction. **The full $GR^3$ design achieves the strongest accuracy–length balance, confirming that the gain stems from the interplay of all three components rather than any single one.**

### 4.3.3. WHY DOES $GR^3$ OUTPERFORM GRPO?

We observe a counterintuitive phenomenon: in many settings, $GR^3$ not only shortens responses but also achieves stronger downstream performance than standard GRPO, while maintaining a positive reward gap relative to the GRPO baseline. We attribute this to a difference in how the optimization signal is structured.

Under unconstrained RL such as GRPO, policies often drift toward over-extended reasoning trajectories. Although these trajectories may eventually reach correct answers, they tend to contain many low-contributing tokens. From an optimization perspective, this spreads the learning signal thinly across long responses, reducing the effective impact of reward on the most important reasoning steps. **By discouraging unnecessary verbosity, $GR^3$ compresses reasoning traces while preserving decisive steps.** This increases the signal density of reward with respect to tokens, allowing optimization to focus more strongly on causally important reasoning patterns rather than distributing gradients over verbose but weakly relevant tokens. Qualitative rollout examples are provided in Appendix F.

## 5. Related Work

Despite remarkable progress, reinforcement learning (Kaelbling et al., 1996; Bai et al., 2022; Guo et al., 2025) suffers from high inference costs and growing generation lengths, a bottleneck we term length inflation.

One line of work studies efficient reasoning (Feng et al., 2025; Sui et al., 2025), aiming to improve the accuracy–cost trade-off of long chain-of-thought models. Early approaches rely on prompting or supervised fine-tuning to encourage shorter reasoning traces (Ma et al., 2025a; Xia et al., 2025; Ma et al., 2025b). More recent methods apply RL to directly optimize efficiency via length-aware objectives (Arora & Zanette, 2025; Liu et al., 2025c;b; Yu et al., 2025) or rejection sampling (Shrivastava et al., 2025). While effective at reducing token usage, such approaches can degrade performance or introduce brittle optimization dynamics due to poorly calibrated penalties or shortcut solutions (Cheng et al., 2025).

Another line of work attributes length inflation in RLHF to reward hacking and length bias (Skalse et al., 2022; Gao et al., 2023; Singhal et al., 2023). Because reward models may implicitly favor longer responses, verbosity can arise from exploiting reward artifacts rather than true capability gains (Shen et al., 2023). Prior efforts mitigate this by improving reward modeling and calibration (Chen et al., 2024a; Wang et al., 2025) or by applying post-hoc reward correction (Huang et al., 2024), though many of these solutions are tailored to specific training settings.

Our method is most closely related to RL-based efficient reasoning and length-regularized policy optimization. We propose $GR^3$, which controls length through group-relative reward rescaling and improves the performance–cost Pareto frontier without sacrificing task performance. While multiplicative reward formulations have appeared in prior studies (Lu et al., 2023; He et al., 2025), our focus is not multiplication itself, but its integration with group-relative regularization and advantage-aware calibration under an advantage-level view of reward shaping, as detailed in Section 3.

## 6. Conclusion

In this work, we identify length inflation as a fundamental inefficiency in RL-trained LLMs, where models tend toward unnecessary verbosity or overthinking. We propose Group Relative Reward Rescaling ($GR^3$), a general framework for lossless length control that regulates reasoning length through a multiplicative, group-relative formulation with advantage-aware calibration. Across both RLVR and RLHF settings, $GR^3$ consistently shifts the performance–cost Pareto frontier outward, reducing token usage while preserving or even improving model capability. These results show that verbosity is not a prerequisite for intelligence, and position $GR^3$ as a practical and general paradigm for training efficient, high-performing LLMs.

## Impact Statement

This paper presents a method to mitigate length inflation in Large Language Models trained via Reinforcement Learning. The primary positive impact of this work lies in promoting computational efficiency and environmental sustainability ("Green AI"). By significantly reducing token generation, e.g., saving over 40% of tokens in reasoning tasks without compromising performance, $GR^3$ directly contributes to lowering the financial costs, inference latency, and energy consumption associated with deploying large-scale reasoning models.

Furthermore, this work addresses the alignment challenge of reward hacking, where models exploit verbosity to maximize rewards rather than improving genuine capability. By decoupling performance gains from unnecessary length, we facilitate the development of more concise, interpretable, and user-aligned AI systems. Nevertheless, practitioners should monitor for potential over-truncation in safety-critical domains where exhaustive reasoning traces are essential for verification.

## Acknowledgements

We sincerely thank the reviewers for their insightful comments and valuable suggestions. This work was supported by the Natural Science Foundation of China (No. 62536008, 62306303, 62476265).

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

## A. Connection to Heuristic Gating Mechanisms

In Section 3.1, we motivated multiplicative shaping primarily through the lens of removing the compensatory optimization shortcut inherent in additive shaping. In this section, we provide an alternative perspective by analyzing the relationship between our approach and heuristic gating mechanisms (Arora & Zanette, 2025; Cheng et al., 2025). We demonstrate that multiplicative shaping can be viewed as a principled generalization of heuristic gating: it mathematically reduces to gating in binary reward settings while providing a robust, "soft" gating mechanism in continuous reward landscapes where hard indicators fail.

**Equivalence in Binary Reward Settings.**   Heuristic gating is a common enhancement to additive shaping in efficient reasoning (RLVR), which prevents models from optimizing length at the expense of accuracy. It typically employs an indicator function $\mathbb{I}(R = 1)$ to apply length penalties only when the response is correct.

Let $P$ denote a generic length-based penalty term (e.g., a negative function of length). Standard *gated additive shaping* modifies the shaping term $S$ in Eq. 5 to be conditional on task success:

$$\textbf{Gated Additive:} \quad \hat{R}^{(+,g)} = R + \lambda \cdot S_{\text{gate}},$$
$$\text{where} \quad S_{\text{gate}} = \mathbb{I}(R = 1) \cdot P.$$

Here, $\mathbb{I}(R = 1)$ acts as a hard gate, $R \in \{0, 1\}$ is the binary task outcome.

Now, consider our *multiplicative shaping* defined in Eq. 7:

$$\textbf{Multiplicative:} \quad \hat{R}^{(\times)} = R \cdot S_{\text{mult}}.$$

To facilitate comparison, we can decompose the scaling factor $S_{\text{mult}}$ into a baseline and a deviation term. We rewrite $S_{\text{mult}} = 1 + (S_{\text{mult}} - 1)$, where the deviation corresponds to the implicit penalty applied by the scaling mechanism:

$$\lambda P := S_{\text{mult}} - 1 \quad \implies \quad S_{\text{mult}} = 1 + \lambda P.$$

We analyze the behavior in the two binary states:

- **Case $R = 0$ (Failure):**

$$\hat{R}^{(+,g)} = 0 + \lambda \cdot (0 \cdot P) = 0$$
$$\hat{R}^{(\times)} = 0 \cdot (1 + \lambda P) = 0$$

  Both methods deactivate the penalty, preventing premature termination on hard instances.

- **Case $R = 1$ (Success):**

$$\hat{R}^{(+,g)} = 1 + \lambda \cdot (1 \cdot P) = 1 + \lambda P$$
$$\hat{R}^{(\times)} = 1 \cdot (1 + \lambda P) = 1 + \lambda P$$

  Both methods apply the full penalty to incentivize efficiency among correct solutions.

Thus, in the strict binary reward setting typical of RLVR, multiplicative shaping is mathematically equivalent to heuristic gating. It inherits the desirable property of protecting the policy from being penalized on incorrect reasoning paths.

**Generalization to Continuous Rewards.**   The limitation of heuristic gating becomes apparent when transitioning to continuous reward settings, such as RLHF (where rewards are typically given by a reward model) or reasoning tasks with partial credit.

In these scenarios, the hard indicator $\mathbb{I}(R = 1)$ is ill-defined. Naively replacing it with a threshold $\mathbb{I}(R > \tau)$ introduces hyperparameters and optimization discontinuities. Conversely, removing the gate entirely (reverting to pure additive shaping) reintroduces the trade-off discussed in Proposition 3.1, where the model can improve $\hat{R}^{(+)}$ by shortening length even if $R$ degrades slightly.

Multiplicative shaping solves this by acting as a soft gating mechanism. As derived in Proposition 3.2, the group-normalized advantage under multiplicative shaping contains the following term governing the length signal:

$$A(\hat{R}^{(\times)}) \propto R \cdot (S - \mu_S) + \dots$$

This decomposition demonstrates that the impact of length variation $(S - \mu_S)$ on the advantage is explicitly scaled by the task reward $R$. This creates a dynamic reweighting of the learning signal:

- **Low Quality** ($R \approx 0$): The length signal is suppressed ($R \cdot (S - \mu_S) \approx 0$). The advantage is dominated by the need to improve task correctness, and the policy receives little to no signal regarding length. This mimics an inactive gate, preventing the model from collapsing to short but incorrect responses.

- **High Quality** ($R \approx 1$): The length signal is fully active ($R \cdot (S - \mu_S) \approx S - \mu_S$). The advantage significantly favors shorter trajectories within the successful group. This mimics an active gate, effectively prioritizing efficiency once capability is secured.

This property effectively interpolates between "no penalty" and "full penalty" based on the response quality. Consequently, GR$^3$ allows us to apply strong length regularization in RLHF without the risk of the model collapsing to short, low-quality responses, as evidenced by the dynamics in Figure 2.

**Empirical Observation.** We further conduct an analytical experiment, illustrated in Figure 5. Specifically, we convert our multiplicative shaping term into a penalty form used in gated additive shaping by defining $\lambda P := S_{\text{mult}} - 1$. We then introduce different thresholds $\mathbb{I}(R > \tau)$ to extend gated additive shaping to continuous-reward RLHF settings. We observe that, due to optimization discontinuities, all choices of $\tau$ lead to performance that falls short of standard GRPO. At the same time, generation length is reduced more aggressively, dropping below the typical level of the base policy model.

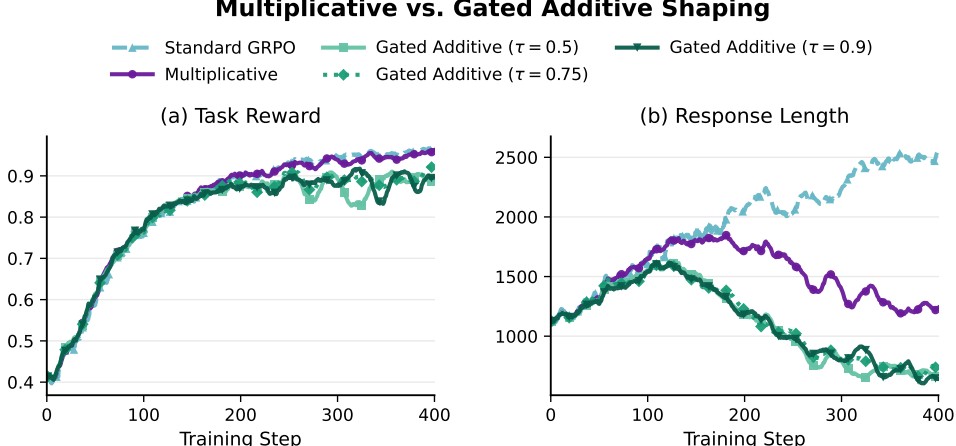

*Figure 5.* Comparison of multiplicative and gated additive shaping during RLHF training. Multiplicative shaping matches standard GRPO in task reward while achieving controlled length reduction. In contrast, gated additive variants with different reward thresholds underperform and produce overly short responses, reflecting optimization instability introduced by hard gating.

## B. Analysis of the Difficulty Over-Adaptation Phenomenon

In Section 3.2, we discussed how group-relative length regularization adapts the length budget to on-policy statistics. While this removes the rigidity of global thresholds, we observe an unintended side effect in certain shaping strategies (e.g., Efficiently (Arora & Zanette, 2025)): the policy can become *over-adaptive* to perceived task difficulty, a phenomenon we term *difficulty over-adaptation*. Concretely, the model tends to aggressively compress reasoning on easy prompts while failing to effectively restrain excessive length on hard ones, as reflected in Table 2 and illustrated by the examples in Table 6. In other words, the regularizer distorts how reasoning effort is allocated across difficulty levels.

*Table 6.* Example output from the Efficiently (Arora & Zanette, 2025) 1.5B baseline at step 800, showing excessive reasoning truncation and omission of critical intermediate steps, resulting in an incoherent chain of thought.

---

**Prompt**
A $90°$ rotation around the origin in the counter-clockwise direction is applied to $7 + 2i$. What is the resulting complex number? Let's think step by step and output the final answer within `\boxed{}`.

---

```
<think>
7cos90 -2i sin90=0-2i
```
                                             // Extremely short reasoning performed
```
</think>
```

$$\boxed{-2i}$$

                                             // Fails to present the correct answer.

---

To understand the mechanism behind this skew, we analyze the sensitivity of the reward function to changes in length. Consider the Efficiently (Arora & Zanette, 2025) formulation:

$$\hat{R}(x, y^{(i)}) = R(x, y^{(i)}) - \lambda \cdot \mathbb{I}(R(x, y^{(i)}) = 1) \cdot s\left(\frac{\ell^{(i)} - \bar{\ell}}{\sigma_\ell}\right).$$

In this formulation, the input to the sigmoid function $s(\cdot)$ is amplified by the factor $1/\sigma_\ell$. This means that the marginal penalty associated with increasing the response length is inversely proportional to the statistical dispersion ($\sigma_\ell$) of the group.

This dependency creates instability across difficulty regimes. On easier prompts, the policy is typically confident and converges to consistent responses, causing the length standard deviation to collapse (i.e., $\sigma_\ell \to 0$). Consequently, the scaling factor $1/\sigma_\ell$ becomes extremely large. In this low-variance regime, even a deviation of a single token is treated as a massive statistical outlier, triggering a severe drop in reward. This hypersensitivity forces the model to over-compress simple responses to avoid harsh penalties. Conversely, on harder prompts, the policy often explores diverse reasoning paths, resulting in a larger $\sigma_\ell$. This attenuates the penalty signal, allowing longer generations to persist with relatively little cost.

In contrast, GR$^3$ normalizes the penalty based on the characteristic scale (mean length $\bar{\ell}$) rather than dispersion:

$$\hat{R}(x, y^{(i)}) = R(x, y^{(i)}) \cdot \frac{1}{1 + \alpha \cdot \frac{\ell^{(i)}}{\bar{\ell}}}.$$

Here, the sensitivity of the penalty depends on the ratio relative to $\bar{\ell}$. While the mean length $\bar{\ell}$ is naturally smaller for easier tasks (appropriately making the budget tighter), it represents the physical scale of the response and does not collapse to near-zero values as the model becomes confident. By normalizing based on scale rather than variance, GR$^3$ provides a stable regularization signal that remains robust to the model's convergence state, effectively mitigating the imbalance in compression pressure across difficulty levels.

## C. The Dilemma of High Reward Density

In this section, we analyze a fundamental structural tension that arises when length regularization is combined with group-normalized advantages. We show that under *high reward density*, a seemingly desirable strict requirement—ensuring that all highest-reward trajectories retain positive advantage—is in general mathematically infeasible. We then show that even a relaxed average-case criterion can degenerate in the limiting case where all sampled trajectories achieve $R_{\max}$, as a consequence of the convexity of multiplicative length rescaling. These observations motivate the relaxed, calibration-based constraint and the online filtering strategy adopted in Section 3.3.

### C.1. Impossibility of the Strict Advantage-Preservation Objective

**A Strict but Natural Objective.** A natural objective for length-aware reinforcement learning is to preserve the optimization signal of the best solutions. Concretely, consider the following strict condition: all trajectories achieving the maximum task reward $R_{\max}$ within a sampled group should receive positive advantages after length regularization. Intuitively, these

trajectories represent the highest-quality responses, and assigning them negative advantages risks discouraging correct reasoning behaviors.

Under GRPO-style normalization, the strict objective is therefore equivalent to requiring

$$\hat{R}(x, y^{(j)}) > \mu_{\hat{R}}, \quad \forall j \in \mathcal{H},$$

where $\mathcal{H} := \{j : R(x, y^{(j)}) = R_{\max}\}$ denotes the set of highest-reward trajectories, $\hat{R}(x, y^{(j)})$ is the regularized reward and $\mu_{\hat{R}}$ is the mean regularized reward within the group.

**Impracticality Under High Reward Density.** We now show that under high reward density, the strict objective of preserving positive advantages for *all* highest-reward trajectories is mathematically infeasible, regardless of how small the length penalty coefficient $\alpha$ is.

Under GR$^3$, the regularized reward is

$$\hat{R}(x, y^{(j)}) = \frac{R(x, y^{(j)})}{1 + \alpha \cdot \frac{\ell^{(j)}}{\bar{\ell}}}.$$

For all $j \in \mathcal{H}$ we have $R(x, y^{(j)}) = R_{\max}$, so variation in $\hat{R}$ depends only on length. Since the regularization term is monotonically decreasing in $\ell^{(j)}$, the longest trajectory in $\mathcal{H}$,

$$\ell_{\max} = \max_{j \in \mathcal{H}} \ell^{(j)},$$

attains the smallest regularized reward

$$\hat{R}_{\min} = \frac{R_{\max}}{1 + \alpha \cdot \frac{\ell_{\max}}{\bar{\ell}}}.$$

When reward density is high, the group mean $\mu_{\hat{R}}$ is dominated by trajectories in $\mathcal{H}$ and is therefore approximately their average. Since the minimum of a set cannot exceed its mean, we must have $\hat{R}_{\min} \leq \mu_{\hat{R}}$. Hence, at least one highest-reward trajectory receives a non-positive advantage. This shows that, under high reward density, it is impossible for *all* highest-reward trajectories to maintain positive advantages after group normalization; the conflict is structural rather than a consequence of hyperparameter choice.

This impossibility persists even in the limit $\alpha \to 0$. Using a first-order expansion,

$$\hat{R}(x, y^{(j)}) \approx R_{\max} \cdot \left(1 - \alpha \cdot \frac{\ell^{(j)}}{\bar{\ell}}\right),$$

and denoting by

$$\bar{\ell}_{\mathcal{H}} = \frac{1}{k} \sum_{j \in \mathcal{H}} \ell^{(j)},$$

the mean length among highest-reward trajectories, we obtain

$$\hat{R}(x, y^{(j)}) - \mu_{\hat{R}} \approx -R_{\max} \cdot \alpha \cdot \frac{\ell^{(j)} - \bar{\ell}_{\mathcal{H}}}{\bar{\ell}}.$$

Hence

$$\hat{A}^{(j)} \propto -\left(\ell^{(j)} - \bar{\ell}_{\mathcal{H}}\right).$$

so any highest-reward trajectory longer than the mean length of $\mathcal{H}$ must receive a negative advantage, no matter how small $\alpha$ is. Group normalization enforces a zero-mean constraint that inevitably induces sign flips among equally correct trajectories with different lengths.

**Empirical Observation.** Figure 6 empirically illustrates this phenomenon. The horizontal axis shows the proportion of trajectories in a group achieving $R_{\max}$ (reward density), and the vertical axis shows the fraction of highest-reward trajectories that satisfy the strict condition $A_i > 0$. Even with a very small penalty strength (Figure 6(a), $\alpha = 0.05$), the satisfaction rate declines as reward density increases. With a larger $\alpha$ (Figure 6(b), $\alpha = 5.0$), the decline becomes much more pronounced. These results confirm that violations of the strict condition arise from an inherent structural conflict rather than poor hyperparameter tuning.

This impossibility result directly motivates the relaxed calibration strategy in Section 3.3. Instead of attempting to protect the longest highest-reward trajectory—which is generally infeasible—we adopt an average-case criterion that ensures a typical high-quality trajectory, whose length is near the group mean, remains above the group-average regularized reward. This leads naturally to the practical constraint in Eq. (16).

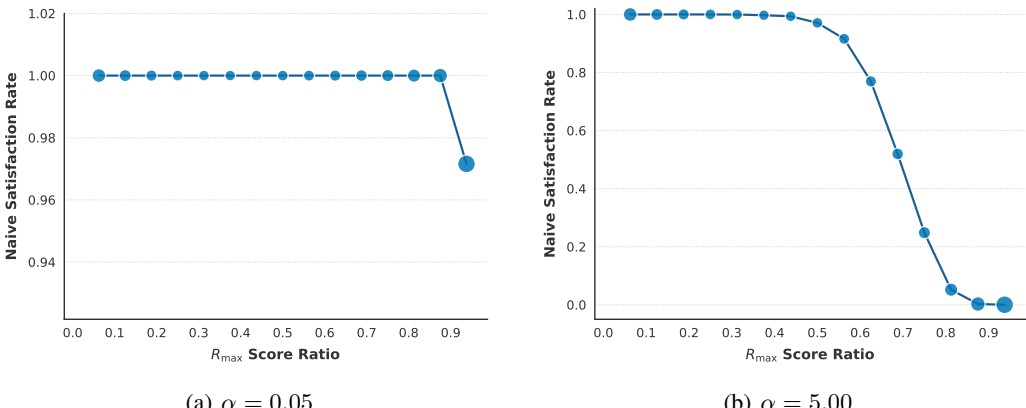

(a) $\alpha = 0.05$          (b) $\alpha = 5.00$

*Figure 6.* Effect of reward density on the strict advantage-preservation objective. As reward density increases, the strict constraint satisfaction rate decreases, even for a small penalty coefficient $\alpha = 0.05$, and more noticeably for a larger penalty $\alpha = 5.0$.

### C.2. Degeneracy of the Average-Case Criterion in the All-$R_{\max}$ Limit

Recall the average-case calibration criterion (Eq. (16)), which requires that a representative high-quality trajectory (reward $R_{\max}$ and average length $\bar{\ell}$) retains non-negative advantage:

$$\frac{R_{\max}}{1 + \alpha} \;\geq\; \mu_{\hat{R}},$$

where $\mu_{\hat{R}}$ denotes the within-group mean of the regularized rewards. As noted in Section 3.3, this constraint can fail in the limiting case where all trajectories in the group achieve $R_{\max}$, motivating online filtering.

**All-$R_{\max}$ case reduces the condition to a Jensen inequality.** Assume a sampled group satisfies $R(x, y^{(i)}) = R_{\max}$ for all $i \in \{1, \ldots, G\}$. Under GR$^3$, the regularized reward is

$$\hat{R}^{(i)} \;=\; R_{\max} \cdot \frac{1}{1 + \alpha \cdot \frac{\ell^{(i)}}{\bar{\ell}}}.$$

Define the normalized length ratio $z_i := \ell^{(i)}/\bar{\ell}$, which obeys $\frac{1}{G} \sum_{i=1}^{G} z_i = 1$. Let

$$f(z) := \frac{1}{1 + \alpha z}, \qquad z \geq 0.$$

Then $\mu_{\hat{R}} = R_{\max} \cdot \frac{1}{G} \sum_{i=1}^{G} f(z_i)$, and Eq. (16) becomes

$$f(1) \;\geq\; \frac{1}{G} \sum_{i=1}^{G} f(z_i).$$

**Convexity flips the inequality.** A direct calculation gives

$$f''(z) = \frac{2\alpha^2}{(1 + \alpha z)^3} > 0,$$

so $f$ is *convex* on $[0, \infty)$. By Jensen's inequality,

$$f\left(\frac{1}{G}\sum_{i=1}^{G} z_i\right) \leq \frac{1}{G}\sum_{i=1}^{G} f(z_i).$$

Using $\frac{1}{G}\sum_i z_i = 1$, we obtain

$$f(1) \leq \frac{1}{G}\sum_{i=1}^{G} f(z_i),$$

which is the *reverse* of the desired condition. Moreover, the inequality is strict whenever the lengths are not all identical (i.e., when $z_i$ is not constant). Therefore, in an all-$R_{max}$ group, the average-case constraint in Eq. (16) can only hold in the degenerate case $\ell^{(i)} = \cdots = \ell^{(G)}$; otherwise it must fail. This explains why we exclude such groups via online filtering in practice.

## D. Detailed Experimental Settings

We evaluate GR$^3$ in representative post-training regimes spanning RLVR and RLHF. In the RLVR setting, we study whether the model can reduce unnecessary long CoT reasoning while maintaining task performance improvement. As for the RLHF setting, we examine whether our method mitigates reward hacking (Amodei et al., 2016) and produces well-aligned responses with natural response lengths.

In this section, we provide the detailed hyperparameters and configurations used in our experiments. All experiments are conducted using veRL (Sheng et al., 2024) as the training framework, based on a standard GRPO advantage estimator. The detailed implementation settings for the experiments are shown in Table 7.

Moreover, we evaluate mathematical reasoning on AIME-24 (MAA, 2024), AIME-25 (MAA, 2025), AMC-23 (MAA, 2023), and MATH500 (Lightman et al., 2023); code generation on LiveCodeBench v6 (Jain et al., 2024) and MultiPL-E (Cassano et al., 2022); and conversational ability on Arena-Hard-Auto (Li et al., 2024) and AlpacaEval (Dubois et al., 2024). Following standard practice, we report the length-controlled (LC) win rate for AlpacaEval. For Arena-Hard-Auto, we do not apply length control and instead use scores derived directly from the original pairwise judge.

*Table 7.* Hyper-parameters used in our experiments.

| Hyperparameter | Math | Code | Chat |
|---|---|---|---|
| GR$^3$ Coefficient $\alpha$ | 0.33 | 0.33 | 0.00133 |
| KL Coefficient $\beta$ | 0.0 | 0.0 | 0.001 |
| Batch Size | 128 | 128 | 128 |
| Mini Batch Size | 128 | 128 | 128 |
| Prompt Length | 1024 | 1024 | 1024 |
| Rollout Length | 16384 | 16384 | 4096 |
| Rollout Temperature | 1.0 | 1.0 | 1.0 |
| Rollout Group Size | 16 | 8 | 8 |
| Optimizer | AdamW | AdamW | AdamW |
| Weight Decay | 0.01 | 0.01 | 0.01 |
| Learning Rate | 2e-6 | 2e-6 | 1e-6 |
| Scheduler Type | Constant | Constant | Constant |
| Training Step | 1000 | 800 | 400 |
| Evaluation Length | 32768 | 32768 | 8192 |
| Evaluation Temperature | 0.6 | 0.6 | 0.0 |

# E. Additional Experimental Results

## E.1. Mathematical Reasoning Results on 1.5B Models

This section presents experimental results on mathematical reasoning tasks, using DeepSeek–R1–Distill–1.5B as the base model. As shown in Table 8, the findings are consistent with the experiments of DeepSeek–R1–Distill–7B: $GR^3$ not only significantly enhances the model's reasoning capabilities but also substantially reduces the length of the generated output in terms of tokens, demonstrating good performance across different scales.

*Table 8.* Mathematical reasoning performance for 1.5B models.

| Model | AIME24 | | AIME25 | | AMC23 | | MATH500 | |
|---|---|---|---|---|---|---|---|---|
| | Avg@32↑ | #Token↓ | Avg@32↑ | #Token↓ | Avg@16↑ | #Token↓ | Avg@4↑ | #Token↓ |
| *Initial model* | | | | | | | | |
| DeepSeek–R1–Distill–1.5B | 30.0 | 16531 | 23.6 | 15799 | 70.8 | 9351 | 83.9 | 5399 |
| *Length-oriented RL* | | | | | | | | |
| LCR1–1.5B | 23.5 | 9071 | 20.6 | 8275 | 67.8 | 4170 | 81.9 | 2520 |
| Laser–DE–L4096–1.5B | 30.1 | 5770 | 24.1 | 5008 | 73.4 | 3110 | 84.6 | 1931 |
| AdaptThink–1.5B | 34.2 | 9204 | 24.7 | 9234 | 63.3 | 2859 | 80.9 | 1649 |
| DLER–R1–1.5B | 34.3 | 3839 | 27.8 | 3153 | 82.1 | 2419 | 87.2 | 1783 |
| *Performance-oriented RL* | | | | | | | | |
| GRPO | 39.6 | 13054 | 31.4 | 12985 | 81.9 | 9917 | 85.6 | 7138 |
| $GR^3$ *(ours)* | 45.2 | 8381 | 32.8 | 8137 | 81.6 | 4153 | 89.3 | 2214 |

## E.2. Ablation Results on Penalty Strength $\alpha$

We analyze the sensitivity of $GR^3$ to the length penalty coefficient $\alpha$ by sweeping its value under identical training settings. Detailed results on 1.5B models are shown in Table 9.

When $\alpha$ is large (e.g., $\alpha = 1.0$), the multiplicative rescaling term heavily penalizes long trajectories, largely independent of their reward level. In this regime, $GR^3$ behaves similarly to conventional length-penalized RL, where optimization is driven primarily by shortening responses rather than improving solution quality. Although token usage is significantly reduced, the performance gains over the initial model become noticeably smaller, indicating that overly aggressive regularization suppresses useful long-form reasoning.

As $\alpha$ decreases, response length increases in a gradual and well-behaved manner, while task performance first improves and then saturates. In particular, moving from $\alpha = 0.33$ to smaller values (e.g., $0.2$ and $0.1$) yields longer generations but only marginal or inconsistent accuracy improvements. This empirical trend aligns closely with the analysis in Section 3.3. Once the penalty is weak enough that the advantage of representative high-quality trajectories is preserved, further reducing $\alpha$ mainly relaxes the length constraint without introducing stronger optimization signals. In other words, the training has already crossed the advantage-preservation boundary, after which additional reasoning length no longer translates into meaningful performance gains.

Overall, $\alpha$ controls distinct behavioral regimes, and $GR^3$ provides a principled way to select it near the advantage-preserving transition between length control and capability gains.

## E.3. Component Ablation and Controlled Baseline Reproduction

We further disentangle multiplicative shaping, the group-relative term, and advantage-aware calibration using a controlled 500-step training and evaluation budget. As shown in Table 10, multiplicative shaping contributes to performance retention, but does not alone explain the full gain; the complete $GR^3$ design provides the strongest aggregate accuracy–length balance.

# F. Qualitative Analysis of Rollout Trajectories

To better understand the behavioral differences induced by the two training objectives, we present representative rollout examples from a $GR^3$-trained model and a GRPO-trained baseline on the same reasoning prompt. Tables 11 and 12 show

*Table 9.* Hyperparameter sweep of the length penalty coefficient $\alpha$ in $GR^3$ on 1.5B models.

| Model | AIME24 | | AIME25 | | AMC23 | | MATH500 | |
|---|---|---|---|---|---|---|---|---|
| | Avg@32↑ | #Token↓ | Avg@32↑ | #Token↓ | Avg@16↑ | #Token↓ | Avg@4↑ | #Token↓ |
| *Initial model* | | | | | | | | |
| DeepSeek–R1–Distill–1.5B | 30.0 | 16531 | 23.6 | 15799 | 70.8 | 9351 | 83.9 | 5399 |
| *Our method: varying $\alpha$* | | | | | | | | |
| $GR^3$ ($\alpha = 1.00$) | 34.9 | 6316 | 27.4 | 5927 | 75.2 | 1754 | 83.0 | 848 |
| $GR^3$ ($\alpha = 0.50$) | 40.5 | 7923 | 29.6 | 7399 | 82.3 | 3245 | 87.1 | 1705 |
| $GR^3$ ($\alpha = 0.33$) | **45.2** | **8381** | **32.8** | **8137** | **81.6** | **4153** | **89.3** | **2214** |
| $GR^3$ ($\alpha = 0.20$) | 45.1 | 10174 | 32.4 | 10250 | 83.0 | 4887 | 88.9 | 2838 |
| $GR^3$ ($\alpha = 0.10$) | 43.2 | 10759 | 31.7 | 10701 | 83.0 | 6235 | 88.2 | 3419 |

*Table 10.* Controlled component ablation on 1.5B models. All models are trained for 500 steps. Rows without multiplicative shaping use additive shaping with $\lambda = 0.01$.

| Multiplicative Shaping | Our Group-Relative Term | Advantage-Aware Calibration | AIME24 | | AIME25 | | AMC23 | | MATH500 | |
|---|---|---|---|---|---|---|---|---|---|
| | | | Acc.↑ | #Token↓ | Acc.↑ | #Token↓ | Acc.↑ | #Token↓ | Acc.↑ | #Token↓ |
| ✓ | ✓ | ✓ | **41.0** | **9089** | **29.2** | **8841** | **81.5** | **4513** | **86.6** | **2612** |
| ✓ | ✓ | ✗ (large $\alpha$) | 32.1 | 5355 | 25.5 | 4667 | 69.3 | 1789 | 82.5 | 845 |
| ✓ | ✓ | ✗ (small $\alpha$) | 40.9 | 12288 | 28.9 | 13115 | 80.0 | 7477 | 87.6 | 4046 |
| ✓ | ✗ (LC-R1 style) | ✗ | 39.7 | 10391 | 28.2 | 10730 | 78.2 | 3254 | 81.6 | 1436 |
| ✓ | ✗ (Efficiently style) | ✗ | 38.5 | 11041 | 29.3 | 11422 | 73.2 | 3944 | 81.2 | 1534 |
| ✗ | ✗ (LC-R1 style) | ✗ | 25.1 | 5120 | 21.2 | 4097 | 68.5 | 2590 | 76.1 | 1488 |
| ✗ | ✗ (Efficiently style) | ✗ | 26.6 | 5313 | 21.3 | 4503 | 67.5 | 2728 | 76.5 | 1506 |

the generation traces.

**$GR^3$: concise reasoning with preserved structure.** As illustrated in Table 11, the $GR^3$-trained model produces a reasoning trajectory that is both structured and economical. The solution follows a clear progression: (i) restating the task, (ii) identifying the periodicity of directions under full rotations, (iii) reducing the angle to a remainder modulo $360°$, and (iv) mapping the residual rotation to a compass direction. Each step contributes directly to advancing the solution, and intermediate checks serve to confirm rather than re-derive earlier results.

Importantly, the trajectory terminates decisively with a correctly formatted boxed answer. The reasoning chain is neither artificially shortened nor overly verbose; instead, redundant re-computation and self-doubt loops are largely absent. This reflects a policy that has learned to allocate tokens primarily to causally relevant steps.

**GRPO: verbose loops and diluted signal.** In contrast, the GRPO-trained baseline (Table 12) exhibits substantially different behavior. Although it repeatedly identifies correct intermediate facts—such as the $360°$ periodicity and the $2250 \bmod 360 = 90°$ reduction—it frequently re-derives them, questions previously established conclusions, and oscillates between equivalent formulations (e.g., full rotations vs. fractional rotations). The trajectory contains multiple self-corrections that do not introduce new information.

This pattern results in long reasoning traces where many tokens are only weakly related to forward progress. From an optimization perspective, such trajectories distribute the reward signal across a large number of low-impact tokens, reducing the effective learning pressure on decisive reasoning steps. Moreover, despite eventually circling back to the correct direction, the model fails to present a clean final answer in the required boxed format, leaving the solution inconclusive.

**Implications for optimization dynamics.** These qualitative differences align with the mechanism discussed in Section 4.3.3. By down-weighting the advantages of unnecessarily long trajectories, $GR^3$ implicitly favors reasoning paths with higher informational density per token. This encourages the policy to preserve essential logical structure while avoiding non-robust reasoning patterns, such as repetitive or self-doubting loops. As a result, the reward signal becomes more concentrated on tokens that correspond to meaningful reasoning transitions, rather than being diluted across verbose yet

*Table 11.* Example output from our GR³-trained 1.5B model, demonstrating robust, clearly structured step-by-step reasoning and a correct final answer in boxed format.

---

**Prompt**

A figure skater is facing north when she begins to spin to her right. She spins 2250 degrees. Which direction (north, south, east or west) is she facing when she finishes her spin? Let's think step by step and output the final answer within \boxed{}.

---

**Example Output from our GR³-trained 1.5B Model**

`<think>`

Okay, so I need to figure out which direction a figure skater is facing after spinning 2250 degrees to her right. She starts facing north. Hmm, let's break this down step by step.

*// Restates the problem and sets up a step-by-step plan.*

First, I know that a full rotation is 360 degrees, so every time she spins 360 degrees, she'll end up facing the same direction she started. So, to find out where she ends up after 2250 degrees, I should probably find the equivalent angle within a single rotation.

*// Recalls the key concept: directions repeat every 360°.*

Let me calculate how many full rotations 2250 degrees is. I'll divide 2250 by 360. Let's see, 2250 ÷ 360. Let me compute that: 360 × 6 is 2160, and 2250 - 2160 is 90. So, 2250 degrees is equivalent to 6 full rotations plus an additional 90 degrees.

*// Performs division to reduce the rotation to full turns plus a remainder.*

So, she spins 6 full circles, which brings her back to the starting direction, facing north. Then she spins an additional 90 degrees to her right. Now, I need to figure out where she ends up after spinning 90 degrees to the right from north.

*// Uses the result: full turns cancel out, leaving only the 90° remainder.*

Let's recall the directions: when facing north, a right turn would be towards east. So, if she spins 90 degrees to the right, starting from north, she would face east.

*// Maps a 90° right turn from north to the compass direction east.*

Wait, let me double-check that. If she's facing north and spins 90 degrees to the right, her new direction is east. Yeah, that makes sense. So after 2250 degrees, she ends up facing east.

*// Verifies the directional reasoning and confirms the conclusion.*

So the answer should be east.
**Final Answer**

$$\boxed{east}$$

`</think>`

The figure skater starts facing north and spins 2250 degrees to her right.

First, we determine the equivalent angle within a single rotation by dividing 2250 by 360. This gives us 6 full rotations plus an additional 90 degrees (since 2250 - 6*360 = 90).

After completing 6 full rotations, the skater is back facing north. She then spins an additional 90 degrees to her right from north. When facing north and spinning 90 degrees to the right, the skater will face east.

$$\boxed{east}$$

*// Formats the final answer as required.*

---

low-contributing segments.

# G. Author Contributions

Zichao Li led the project.

Jie Lou and Yaojie Lu supervised the project.

Fangchen Dong contributed to the experimental implementation.

Zhiyuan Fan contributed by first sketching the prototype of the method during early discussions.

Mengjie Ren contributed to the experiments on code generation tasks.

Hongyu Lin, Xianpei Han, Debing Zhang, Le Sun, and Xing Yu provided supervision and contributed to technical discussions throughout the project.

*Table 12.* Example output from the GRPO-trained 1.5B baseline, exhibiting redundant reasoning loops, repeated self-doubt, and failure to produce a conclusive final answer.

---

**Prompt**

A figure skater is facing north when she begins to spin to her right. She spins 2250 degrees. Which direction (north, south, east or west) is she facing when she finishes her spin? Let's think step by step and output the final answer within \boxed{}.

---

**Example Output from the GRPO-trained 1.5B Model**

`<think>`

Okay, so I have this problem here about a figure skater spinning. Let me try to figure it out step by step. Hmm, she's facing north and starts spinning to her right. She spins 2250 degrees, and we need to find out which direction she's facing after that. The options are north, south, east, or west.

Alright, first, I know that when you spin, your direction changes based on how many degrees you turn. But since she's starting facing north and spinning to her right, I need to figure out how that affects her direction after a certain number of degrees.

Wait, hold on. When you spin to your right, does that mean you're turning clockwise or counterclockwise? I think in the context of facing north, spinning to your right would be clockwise because if you're facing north and turn right, you're going towards east, then south, then west, and back to north. So, right is clockwise.

But maybe I should clarify. If you're facing north and you spin to your right, you're turning clockwise. So, each 90-degree turn would change your direction. So, 90 degrees to the right from north would be east, another 90 degrees would be south, another 90 degrees would be west, and another 90 degrees back to north. That makes sense.

// Repeatedly re-derives the meaning of "spin to her right".

Now, she spins 2250 degrees. Hmm, that's a lot. Let me think about how to break this down. Since a full circle is 360 degrees, maybe I can find out how many full rotations she makes and then the remaining degrees.

So, if I divide 2250 by 360, that will tell me how many full spins she does and what the remainder is. Let me do that: 2250 ÷ 360. Let me compute that.

First, 360 times 6 is 2160, right? Because 360*6 = 2160. Then, 2250 - 2160 is 90. So, 2250 degrees is equal to 6 full rotations (which bring her back to the starting point) plus an additional 90 degrees. So, effectively, she only needs to consider the 90-degree turn after the full rotations.

But wait, hold on. Wait, is it 6 full rotations or 6.25? Wait, no, 2250 divided by 360 is 6.25, because 360*6=2160, and 2250-2160=90, so it's 6 and 90/360, which is 6.25 full rotations. So, 6.25 rotations is the same as 6 full rotations plus 0.25 of a rotation. But since a full rotation brings her back to the starting direction, the 6 full rotations don't change her facing. So, the key is the 0.25 of a rotation, which is 90 degrees.

But wait, hold on. She starts spinning to her right, which is clockwise. So, spinning 90 degrees clockwise from north would make her face east. So, after 2250 degrees, which is equivalent to 6 full rotations plus 90 degrees, she would have effectively turned 90 degrees clockwise from her starting direction.

But wait, hold on. Wait, the problem says she starts spinning to her right. So, does that mean she is spinning clockwise? Because if she's facing north and spins to her right, that's a clockwise spin. So, each 90-degree turn clockwise would change her direction. So, if she spins 90 degrees clockwise, she would face east. 180 degrees clockwise would face her west, and 270 degrees clockwise would face her south. But in this case, she only spins 90 degrees, which is 2250 degrees, which is 6 full rotations plus 90 degrees. Wait, but 2250 degrees is a lot. Let me make sure I'm calculating this correctly. So, 2250 divided by 360 is equal to... let's see, 360 times 6 is 2160, as I had before. 2250 minus 2160 is 90. So, 2250 degrees is 6 full circles plus 90 degrees. So, she's effectively turned 90 degrees.

// Unnecessarily re-checks the same arithmetic multiple times.

But since she's spinning to her right, which is clockwise, a 90-degree turn would make her face east. So, she starts facing north, spins 90 degrees clockwise, so she faces east. Is that right?

Wait, no, hold on. Wait, maybe I'm mixing up the direction of rotation. If she's spinning to her right, that's a clockwise rotation, but in terms of her facing direction, starting north, spinning 90 degrees clockwise would make her face east.

Wait, but let me think again. When you spin to your right, you're turning clockwise. So, if you're facing north, turning right (clockwise) 90 degrees would make you face east. So, 90 degrees clockwise from north is east. So, 2250 degrees is 6 full rotations plus 90 degrees. So, after the full rotations, she's back to north, and then she turns 90 degrees clockwise, so she's facing east.

// Enters a self-doubt loop, questioning previously correct directional reasoning.

. . .

Wait, no, wait. If you add 2250 degrees, but since 360 is a full circle, 2250 degrees is equivalent to 90 degrees. So, starting from north (0 degrees), adding 90 degrees clockwise is east. So, yes, she's facing east.

But wait, hold on, is it 90 degrees or 270 degrees? Wait, no, because she's spinning to her right, which is clockwise, so 90 degrees clockwise from north is east.

Wait, maybe I should draw a diagram in my mind.

. . .

// Fails to present the final answer, leaving the solution inconclusive.

