# OpenReview forum: "Tackling Length Inflation Without Trade-offs: Group Relative Reward Rescaling for Reinforcement Learning"
_ICML.cc/2026/Conference — ICML 2026 regular_

### Official Review · Reviewer_dftw · 2026-02-19

**Soundness:** 3
**Presentation:** 3
**Significance:** 3
**Originality:** 3
**Overall Recommendation:** 4
**Confidence:** 5

**Summary:**

This paper presents GR3 which is a multiplicative reward shaping method for post-training of LLMs with RL to prevent length exploitation. It is motivated by the current form of the solutions which are additive and susceptible to length optimization due to their formulation. Instead, they propose a multiplicative mechanism that creates interdependence between reward and length of the trajectories. They validate the proposed method across reasoning and alignment to show competitive performance while reducing significant token usage.

**Compliance With Llm Reviewing Policy:**

Affirmed.

**Key Questions For Authors:**

* Why is significantly lower \alpha is necessary for RLHF experiments?

* Could you clarify the reasoning behind the quadratic response length behaviour? I find it a bit counterintuitive.

* Given the relation between R and S, should we still do the std normalization in GRPO or drop it?

* You show the average number of tokens for benchmarks, but is there a difference between the solved and unsolved questions? Does the model spend more tokens if it does not get it right despite the question being from an easier benchmark?

* Do you have tag or format reward for the experiments, this might be the reason for the incomplete responses? Is there a similar behaviour for the base model?

**Limitations:**

There is no explicit limitations section, but they are stated throughout the paper.

**Strengths And Weaknesses:**

# Strengths

* The method is easy to understand and theoretically motivated.

* It is adaptive based on the group statistics unlike other methods which are based on some predetermined constants.
* It is a general method and easily integrable with continuous reward settings where the current reasoning-focused mitigations fails to address.

* Limitations of the method are plainly presented such as in the case of complete R_max solutions similar to zero-variance discussion in DAPO.

* Length is significantly reduced while achieving significant performance across different models and tasks.


# Weaknesses

* Evaluation is heavily skewed toward trained dataset distribution. I would love to see how the math-trained GR3 performs at coding tasks, or at gpqa, vice versa for coding models. I believe it is important to observe whether main takeaways would hold for them as well. Moreover, the base-model performance is already strong in all benchmarks and long generations appear to stem from the base model which is probably due to long-cot distillation. Evaluation at a harder benchmark where the base model struggles would be appreciated. Otherwise one could argue that GR3 attempts to compress the base model while preserving/enhancing the main capabilities .  Lastly, there is no capability evaluation for RLHF-trained models which I believe is crucial for alignment tax discussions.


* Even though authors show that long generations are not necessary to achieve strong benchmark scores, I believe that test-time scaling is still critical. I suspect method might prevent models from spending more tokens on harder tasks.

---

> ### Author Rebuttal · Authors · 2026-03-31
>
> Thank you for recognizing the strengths of our work. We address your concerns below.
>
> **C1: Evaluation skewed toward training distribution; harder benchmarks; capability evaluation for RLHF models (alignment tax).**
>
> A1: Thank you for this valuable suggestion. We address the three sub-concerns in order.
>
> **OOD generalization.** To evaluate generalization beyond the training domain, we additionally test all RLVR models on GPQA-Diamond and MMLU-Pro:
> |Model|GPQA-D|MMLU-P|
> |-|-|-|
> |DS-R1-1.5B|31.6|37.3|
> |1.5B Math RL GR$^3$/GRPO|35.2 / 35.4|41.3 / 40.2|
> |1.5B Code RL GR$^3$/GRPO|33.2 / 33.1|39.8 / 39.7|
> |DS-R1-7B|49.3|57.8|
> |7B Math RL GR$^3$/GRPO|50.1 / 46.7|58.8 / 59.1|
> |7B Code RL GR$^3$/GRPO|47.3 / 47.8|59.2 / 58.9|
>
> GR$^3$ preserves OOD capability comparable to GRPO and the base model across all settings.
>
> **Harder benchmarks.** We would like to clarify that AIME24/25 are already highly challenging for 1.5B models. We partition problems by base-model pass rate (over 32 samples) and report per-bin accuracy:
>
> |Pass-rate bin|#Prob|Base|GRPO|GR$^3$|
> |-|-|-|-|-|
> |[0, 0.125)|32|1.7|5.8|7.5|
> |[0.125, 0.5)|11|23.9|43.2|54.8|
> |[0.5, 0.875)|10|64.4|82.2|82.5|
> |[0.875, 1.0]|7|92.9|92.4|96.0|
>
> GR$^3$ matches or outperforms GRPO across all difficulty bins, including the hardest problems.
>
> **Alignment tax.** We observe no substantial alignment tax in our RLHF setting. GR$^3$ is slightly better than the base model on both benchmarks, suggesting that it preserves the capability gains brought by RLHF (e.g., MMLU-Pro from 66.9 to 67.3, and GPQA-Diamond from 47.8 to 50.4). Due to the rebuttal space limit, we will include the complete results in the final version.
>
> **C2: Test-time scaling is critical. Method might prevent models from spending more tokens on harder tasks.**
>
> A2: Thank you for this important point. We fully agree that test-time scaling is critical. However, not all scaling strategies are equally efficient. Our goal is to achieve thinking-optimal scaling of test-time compute (as discussed in Towards Thinking-Optimal Scaling of Test-Time Compute for LLM Reasoning, NeurIPS 2025). Our advantage-aware calibration is designed to ensure efficiency gains do not come at the cost of performance, as demonstrated empirically across all settings.
>
> **C3: Why is significantly lower $\alpha$ necessary for RLHF?**
>
> A3: Thank you for this insightful question. The key reason lies in the fundamentally different reward distributions. In RLVR, binary rewards induce a coarse-grained advantage structure invariant to monotonic rescaling, allowing larger $\alpha$. In RLHF, dense continuous rewards mean small perturbations can flip relative advantage ordering, so the advantage-aware constraint becomes much tighter, yielding smaller calibrated $\alpha$.
>
> **C4: Reasoning behind the quadratic response length behavior?**
>
> A4: The key lies in $R \cdot (S - \mu_S)$ in Eq. 14, where $R$ gates the length-control signal. In RLHF, rewards grow gradually during training, so the gate transitions from weak to strong — first allowing reward-seeking behavior to increase length, then length control takes over, producing the quadratic pattern. In RLVR, the gate is binary ($R \in \{0,1\}$), so length control acts consistently from the start and length decreases monotonically. We will add this clarification to the final version.
>
> **C5: Should we keep or drop std normalization in GRPO?**
>
> A5: The std normalization is orthogonal to GR$^3$. It does not affect our analysis of the multiplicative mechanism, the group-relative term, or the advantage-aware calibration, and can be retained or removed independently. We keep it in all experiments for consistency with standard GRPO.
>
> **C6: Difference between solved/unsolved questions? More tokens on incorrect answers even on easier benchmarks?**
>
> A6: Thank you for this question. We observe a consistent pattern: incorrect responses are substantially longer than correct ones, even on easy benchmarks like MATH500.
>
> |Model|Correct (mean/med)|Incorrect (mean/med)|
> |-|-|-|
> |Base|3465 / 2283|15441 / 13512|
> |GRPO|5356 / 4266|17740 / 16145|
> |GR$^3$|1595 / 1048|7383 / 3814|
>
> This suggests excessive length is associated with reasoning inefficiency rather than problem difficulty. GR$^3$ notably reduces length for both correct and incorrect cases.
>
> **C7: Tag/format reward? Could this cause incomplete responses? Similar behavior in base model?**
>
> A7: We do not use any tag or format reward. We attribute incomplete responses to reasoning inefficiency under the generation budget. Increasing the limit from 30k to 60k consistently reduces the not-completed ratio:
> |Model|Not-completed|
> |-|-|
> |Base 30k|16.3%|
> |GRPO 30k/60k|5.4% / 3.1%|
> |GR$^3$ 30k/60k|2.3% / 1.6%|
>
> The base model has the highest ratio, confirming this is not introduced by our method. GR$^3$ consistently achieves lower not-completed ratios than GRPO, indicating it alleviates rather than exacerbates this issue.
>
> Please let us know if you have any additional questions.

---

> > ### Author Rebuttal · Reviewer_dftw · 2026-04-03
> >
> > My concerns has been adressed, and I maintain my score.

---

> > > ### Author Response · Authors · 2026-04-06
> > >
> > > Thank you for acknowledging our rebuttal efforts. We sincerely appreciate your time and consideration.

---

### Official Review · Reviewer_Wn6N · 2026-03-04

**Soundness:** 3
**Presentation:** 3
**Significance:** 2
**Originality:** 2
**Overall Recommendation:** 4
**Confidence:** 1

**Summary:**

This paper aims to address the length inflation issue in RL for language models. Previous approaches usually use additive reward shaping that adds a length penalty term to control the length, however, the authors argue that the additive reward shaping may mislead the policy model to maximize the additive reward rather that the real task reward. Therefore, this paper proposes a multiplicative shaping reward that rescales the original task reward.

The empirical results show that the proposed method can effectively control the length while maintaining performance.

**Compliance With Llm Reviewing Policy:**

Affirmed.

**Final Justification:**

I am not familiar with topic so that I give a borderline accept with a very low confidence. Please weigh my assessment less and refer to other reviewers' judgement.

**Key Questions For Authors:**

Please see the weaknesses.

**Limitations:**

yes

**Strengths And Weaknesses:**

I am not familiar with the length inflation topic so that I am not able to comment on the contribution or novelty of this paper. In general, this paper studies a real and important problem: length inflation. This paper is easy to follow and well written. The motivation is clear and the additive reward indeed, intuitively, can mislead the policy to maximize the added partial reward. The empirical results look promising. I like the discussion and empirical evidence on the additive reward misleading the policy model not to optimize the real task reward.

However, I have the following questions:

1. There is still a way to hack the current multiplicative reward: all responses are very long, i.e., each $l^{(i)}$ is large and the average $\bar{l}$ is also large. In this case, the multiplicative reward cannot effectively penalize the long sequences because $\frac{l^{(i)}}{\bar{l}}$ is almost 1. During training, the output distribution of the language model will become more deterministic, that is to say the generated response may become less diverse. How the current design ensures that the output responses are all short rather that all long given that these two situations can have same $\frac{l^{(i)}}{\bar{l}}$.

2. The paper claims that additive reward has a trade-off in terms of the weight $\lambda$ and the multiplicative reward eliminates this trade-off. However, I think the current formula of the multiplicative reward also has a trade-off induced by the coefficient $\alpha$.

3. If the generated responses are of different length, due to the length scaling factor, it is inevitable that the reward becomes different from the original accurate correctness reward. How to make sure that the policy still optimizes the original reward objective (e.g., answer correctness)? I guess this may be related to the tuning of $\alpha$?

---

> ### Author Rebuttal · Authors · 2026-03-31
>
> Thank you for your recognition of the strengths of our work. We address your concerns as follows.
>
> **C1: If all responses become uniformly long, multiplicative scaling fails to penalize length.**
>
> A: Thank you for this insightful question. We agree that if all responses within a group were uniformly long, then $l^{(i)} / \bar{l} \approx 1$, and the explicit length modulation would become weak. However, this case is unlikely and unstable in practice.
>
> Empirically, due to RL sampling stochasticity, responses to the same prompt do not collapse to identical lengths. We quantify this using the intra-group coefficient of variation (CV) of response lengths during GR$^3$ training. While the average response length decreases substantially from $8017.25$ at step 1 to $4353.68$ at step 1000, the average CV remains consistently around $0.30$ throughout training, indicating persistent within-group length diversity rather than length collapse.
>
> Mechanistically, GR$^3$ only requires non-zero within-group length variation to induce optimization pressure toward shorter responses. When responses receive the same task reward, a shorter one obtains a more favorable normalized scaling than a longer one, as long as their lengths are not exactly equal. Therefore, even small length differences lead GR$^3$ to favor shorter responses under equal task reward, pushing the policy toward length reduction rather than a uniformly long regime.
>
> Overall, the “all responses are long and nearly identical” scenario is not a stable training outcome: it is both empirically uncommon under stochastic RL sampling and dynamically broken by the relative preference induced by GR$^3$. We will incorporate this analysis and the corresponding statistics into the final version for clarity.
>
> **C2: The current formula of the multiplicative reward also has a trade-off induced by the coefficient alpha.**
>
> A2: Thank you for raising this point. We would like to clarify that the “trade-off” we refer to is not merely the existence of a tunable coefficient, but the empirical phenomenon observed in prior methods, where improving efficiency (i.e., shorter responses) often leads to degraded performance.
>
> In contrast, our method is designed to mitigate this efficiency–performance trade-off in practice. While the coefficient $\alpha$ could, in principle, introduce a trade-off, our key design is that $\alpha$ is not treated as a freely tuned hyperparameter. Instead, it is determined through the proposed advantage-aware calibration, which enforces a constraint that preserves the relative structure of advantages. This ensures that the modified optimization signal remains aligned with the original reward.
>
> As a result, rather than requiring heuristic tuning of $\alpha$ to balance efficiency and performance, our method provides a more principled and stable way to set $\alpha$, significantly reducing sensitivity to this parameter. Empirically, this leads to consistent improvements in efficiency without sacrificing performance. We will provide additional clarification on this point in the final version.
>
> **C3: How to make sure that the policy still optimizes the original reward objective (e.g., answer correctness)? I guess this may be related to the tuning of $\alpha$?**
>
> A3: Thank you for this important question. Yes, this is closely related to how $\alpha$ is handled, and this is exactly why we introduce the advantage-aware calibration step.
>
> The key point is that GR$^3$ does not replace the original task reward with an independent length objective. Instead, it applies a multiplicative rescaling, $\hat{R} = R \cdot S$, where the length factor $S$ only modulates the original reward $R$. Therefore, correctness-related reward remains the foundation of optimization: if a response receives low task reward (e.g., due to an incorrect answer), shortening it alone cannot make it highly rewarded. In this sense, length control is only a reward-dependent adjustment, rather than a new optimization target that overrides correctness.
>
> More importantly, we explicitly control $\alpha$ to avoid distorting the original optimization direction. We do not treat $\alpha$ as a freely tuned coefficient; instead, we determine it through advantage-aware calibration, which is designed to preserve the relative advantage structure induced by the original reward. Intuitively, this prevents the shaped reward from excessively altering the optimization role implied by the original advantages, thereby keeping the policy update aligned with correctness-oriented optimization.
>
> This is also consistent with our empirical results: as shown in Figure 2 of the main paper, GR$^3$ substantially reduces response length while maintaining reward improvement comparable to standard GRPO, rather than achieving brevity by sacrificing task performance. We will clarify this motivation and the role of advantage-aware calibration more explicitly in the final version.
>
> Please let us know if you have any additional questions.

---

> > ### Author Rebuttal · Reviewer_Wn6N · 2026-04-02
> >
> > Thank you for the response. I will keep the positive rating.

---

> > > ### Author Response · Authors · 2026-04-06
> > >
> > > Thank you for maintaining the positive rating. We sincerely appreciate your time and consideration.

---

### Official Review · Reviewer_jLVe · 2026-03-12

**Soundness:** 3
**Presentation:** 2
**Significance:** 3
**Originality:** 2
**Overall Recommendation:** 4
**Confidence:** 4

**Summary:**

This paper studies the problem of length inflation in RL-trained LLMs, where models become increasingly verbose during training. It proposes Group Relative Reward Rescaling (GR3), a length-control method that replaces additive length penalties with a multiplicative reward rescaling scheme, so that the original task reward is downweighted according to the response length. Beyond the multiplicative formulation, the method includes two additional components: group-relative length regularization, which normalizes each response length by the average length within the sampled group, and advantage-aware calibration, which chooses the penalty strength to avoid suppressing strong trajectories too aggressively. The paper argues that these components together provide a more adaptive and stable way to control verbosity than fixed thresholds or uncalibrated penalties. The authors evaluate GR3 on both RLVR tasks such as mathematical reasoning and code generation, and on RLHF chat alignment settings, with the goal of improving the performance–cost tradeoff. Empirically, the paper reports that GR3 can substantially reduce output length while preserving, and in some cases improving, downstream task performance relative to standard GRPO-style training and prior length-control baselines.

**Compliance With Llm Reviewing Policy:**

Affirmed.

**Final Justification:**

My final recommendation is Weak Accept. The paper addresses a practically relevant problem, controlling length inflation in RL-trained LLMs, and the proposed GR3 recipe appears useful in practice. While I still view the paper’s originality as more limited than initially presented, especially since multiplicative reward modifications have prior precedent in LLM RL, I think the work still offers a meaningful contribution through its particular combination of multiplicative rescaling, group-relative regularization, and calibration, together with broad empirical evaluation.

The rebuttal positively changed my evaluation. In particular, the additional ablations and baseline reproduction substantially addressed my main soundness concerns by clarifying that the gains come from the integrated GR3 design rather than from multiplicative shaping alone, and by making the empirical claims more convincing. My remaining reservation is mainly about originality and positioning rather than technical correctness: I think the final version should more carefully contextualize the work relative to prior multiplicative reward-design approaches and present the novelty in a more precise way. Overall, after considering both the paper and the rebuttal, I find the submission technically solid enough and sufficiently significant to merit acceptance.

**Key Questions For Authors:**

1. The appendix states that under strict binary rewards, the multiplicative formulation becomes mathematically equivalent to heuristic gating. Given this, could the authors clarify why GR3 still shows noticeably stronger results on mathematical RLVR benchmarks than competing methods? In particular, which part of the gain should be attributed to the multiplicative form itself, as opposed to the broader training recipe?

2. Relatedly, it would strengthen the paper to provide a more complete ablation that separates the contribution of (i) multiplicative reward rescaling, (ii) group-relative length regularization, and (iii) advantage-aware calibration.

3. The use of open-sourced checkpoints from prior efficient-reasoning methods is understandable given the cost of RL training, but I wonder whether the authors could provide at least one or two more closely matched baseline comparisons trained under the same framework and hyperparameters. This would help rule out confounding factors from differences on training recipe or infra, especially for the most relevant competing baselines.

**Limitations:**

yes

**Strengths And Weaknesses:**

My main concern is originality. The paper strongly positions “replacing additive reward shaping with multiplicative reward shaping” as a central novelty, but multiplicative reward design has already appeared in prior LLM-RL literature (e.g., "Inference-Time Policy Adapters (IPA): Tailoring Extreme-Scale LMs without Fine-tuning", "Rewarding the Unlikely: Lifting GRPO Beyond Distribution Sharpening"). This does not make the present paper uninteresting, but it does weaken the impression that multiplicative reward shaping itself is a new idea in LLM RL; the more credible novelty here seems to be the specific recipe and its application to length control.

On soundness, my main concern is about the use of open-sourced checkpoints of existing efficient-reasoning methods as baselines. Using public checkpoints is understandable given the cost of RL training, and I do not think this is inherently inappropriate. However, because the paper makes fairly strong claims about the superiority of its reward design, comparisons based partly on released checkpoints may conflate method differences with differences in training recipe and infra. I would have found the paper more convincing with at least one or two matched retraining baselines under the same framework and hyperparameter budget for the most relevant competing methods.

The presentation would be improved by explicitly discussing prior LLM-RL work that already uses multiplicative reward modifications, rather than framing the literature almost entirely through additive baselines. I also have two minor clarity issues. First, in the Figure 2 caption, the phrase “the two settings” is slightly ambiguous on first read; it only becomes clear in Section 4.Experiments that these refer to RLVR and RLHF. Second, in some equations the \bar{l} in the denominator is visually hard to parse because the bar nearly overlaps with the fraction line; this is minor, but it did slow reading.

On significance, I think the paper is addressing an important problem, and even an incremental but reliable improvement in this space could be useful to practitioners training reasoning models.

---

> ### Author Rebuttal · Authors · 2026-03-31
>
> Thank you for your recognition of the strengths of our work. We address your concerns as follows.
>
> **C1: My main concern is originality ... multiplicative reward design has already appeared in prior LLM-RL literature.**
>
> A1: Thank you for the insightful comment. We agree that multiplicative forms have appeared in prior LLM-RL works, and we will revise the paper to better acknowledge them. However, our contribution lies not in the use of multiplication per se, but in providing a principled understanding of reward shaping from the perspective of advantage optimization.
>
> Concretely, we show that additive shaping can introduce an independently optimizable shortcut, leading to compensatory effects that ultimately degrade performance. In contrast, multiplicative shaping acts as a reward-dependent rescaling that preserves the structure of advantages while modulating their magnitude, thereby offering a more stable mechanism for controlling auxiliary factors without distorting the core learning signal.
>
> This leads to a general guideline: reward shaping should preserve the structure of advantages rather than introduce distortions, with length control serving as one concrete instantiation. We will clarify this positioning in the revision and better contextualize our contribution alongside prior work, acknowledging their important contributions and outlining the perspective we aim to add.
>
> **C2: On soundness, my main concern is about the use of open-sourced checkpoints of existing efficient-reasoning methods as baselines.**
>
> A2: Thank you for the constructive feedback. To address this concern, during the rebuttal phase we re-trained DLER using its official implementation and aligned the method-agnostic settings with our training recipe, e.g., batch size, learning rate, training steps. The results are summarized below:
> |Method|AIME 2024 Acc.|AIME 2024 Len.|AIME 2025 Acc.|AIME 2025 Len.|AMC 2023 Acc.|AMC 2023 Len.|MATH-500 Acc.|MATH-500 Len.|Avg. Acc.|Avg. Len.|
> |-|-|-|-|-|-|-|-|-|-|-|
> |GRPO|39.5|13305|30.5|12905|81.0|9177|87.6|5860|59.7|10312|
> |DLER|26.5|3442|22.0|3098|74.3|2221|84.5|1534|51.8|2573|
> |GR$^3$|41.0|9089|29.2|8841|81.5|4513|86.6|2612|59.6|6264|
>
> The reproduced results are consistent with the findings in our paper. We will include additional baseline reproductions in the final version.
>
> **C3: Minor clarity issues: (1) “the two settings” in Figure 2 is ambiguous (only later clarified as RLVR vs. RLHF); (2) the \bar{l} in some denominators is visually hard to parse due to overlap with the fraction bar.**
>
> A3: Thank you for pointing out these clarity issues. We will revise the Figure 2 caption to explicitly specify the two settings (RLVR and RLHF) and improve the denominator typography for better readability.
>
> **C4: Clarify why GR$^3$ outperforms competing methods on RLVR benchmarks, and disentangle the gains from multiplicative rescaling vs. other components via more complete ablations.**
>
> A4: Thank you for this important question. We agree that the performance gains of GR$^3$ should not be attributed to multiplicative shaping alone. Rather, GR$^3$ works as an integrated design in which multiplicative shaping provides a unified mechanism for length control, while the group-relative term and advantage-aware calibration are introduced to better preserve optimization quality and reduce performance degradation.
>
> To better disentangle the role of each component, we provide additional ablation results:
> |Multiplicative Shaping|Our Group-Relative Term|Advantage-Aware Calibration|AIME 2024 Acc.|AIME 2024 Len.|AIME 2025 Acc.|AIME 2025 Len.|AMC 2023 Acc.|AMC 2023 Len.|MATH-500 Acc.|MATH-500 Len.|Avg. Acc.|Avg. Len.|
> |-|-|-|-|-|-|-|-|-|-|-|-|-|
> |✓|✓|✓|41.0|9089|29.2|8841|81.5|4513|86.6|2612|59.6|6264|
> |✓|✓|✗ ($\alpha$ too large)|32.1|5355|25.5|4667|69.3|1789|82.5|845|52.4|3164|
> |✓|✓|✗ ($\alpha$ too small)|40.9|12288|28.9|13115|80.0|7477|87.6|4046|59.3|9232|
> |✓|✗ (LC-R1 style)|✗|39.7|10391|28.2|10730|78.2|3254|81.6|1436|56.9|6453|
> |✓|✗ (Efficiently style)|✗|38.5|11041|29.3|11422|73.2|3944|81.2|1534|55.6|6985|
> |✗|✗ (LC-R1 style)|✗|25.1|5120|21.2|4097|68.5|2590|76.1|1488|47.7|3324|
> |✗|✗ (Efficiently style)|✗|26.6|5313|21.3|4503|67.5|2728|76.5|1506|48.0|3513|
>
> These results support two main observations. First, multiplicative shaping is useful, but it is not sufficient by itself to explain the full gain. Second, the best overall trade-off comes from the full GR$^3$ design, where multiplicative shaping, our group-relative term, and advantage-aware calibration work together to preserve performance while controlling output length.
>
> Please let us know if you have any additional questions.
>
> [1] Cheng Z, Chen D, et al. "Optimizing Length Compression in Large Reasoning Models." arXiv:2506.14755.
>
> [2] Arora D, Zanette A. "Training language models to reason efficiently." NeurIPS 2025.
>
> [3] Liu S Y, Dong X, et al. "DLER: Doing Length pEnalty Right-Incentivizing More Intelligence per Token via Reinforcement Learning." arXiv:2510.15110.

---

> > ### Author Rebuttal · Reviewer_jLVe · 2026-04-03
> >
> > Thank you for the rebuttal. The new ablation and baseline reproduction meaningfully strengthen the paper and largely address my main soundness concerns. My remaining issue is mainly about originality/positioning rather than missing experiments. I appreciate the clarification that the contribution is not multiplication per se, and I think this concern would be largely resolved if the final version more carefully positions the work relative to prior multiplicative reward-design approaches in LLM RL.

---

> > > ### Author Response · Authors · 2026-04-06
> > >
> > > Thank you for your constructive feedback and for acknowledging our rebuttal efforts. We are very glad that the new ablations and baselines have resolved your concerns regarding soundness.
> > >
> > > We completely agree with your point on positioning. We commit to carefully contextualizing our work relative to prior multiplicative reward-design approaches in the final version. Specifically, we will expand our related work and discussion sections to clearly clarify our contributions regarding providing a principled understanding of reward shaping from the perspective of advantage optimization.
> > >
> > > Thank you again for the helpful feedback. We appreciate the opportunity to improve the paper's framing.

---

### Official Review · Reviewer_PGQs · 2026-03-13

**Soundness:** 2
**Presentation:** 3
**Significance:** 3
**Originality:** 3
**Overall Recommendation:** 3
**Confidence:** 4

**Summary:**

The paper proposes Group Relative Reward Rescaling (GR³), a method to mitigate length inflation in RL-trained LLMs. Instead of additive length penalties, GR³ multiplies the task reward by a group-relative scaling factor that penalizes longer responses relative to the within-group mean. This is complemented by advantage-aware calibration for principled selection of the penalty strength. Experiments across math reasoning, code generation, and RLHF chat show that GR³ matches or improves standard GRPO performance while significantly reducing token usage.

**Compliance With Llm Reviewing Policy:**

Affirmed.

**Key Questions For Authors:**

Can the authors provide an ablation study that cleanly separates the contributions of multiplicative shaping and group-relative normalization?

**Limitations:**

yes

**Strengths And Weaknesses:**

**Strengths**
- The paper evaluates across three task domains (math, code, chat), two model scales (1.5B, 7B), and multiple benchmarks per domain. The consistent pattern of maintaining or improving performance while substantially reducing tokens is very convincing.

- The "increase-then-decrease" length pattern in RLHF (Figure 2) is compelling — it shows GR³ adaptively allocates reasoning budget rather than merely truncating responses.

- The advantage-aware calibration (Section 3.3) provides a grounded protocol for choosing α via Constraint Satisfaction Rate, avoiding ad-hoc tuning.

&nbsp;

**Weaknesses**
As the paper itself notes (Appendix A), in binary reward settings typical of RLVR, gated additive shaping is mathematically equivalent to multiplicative shaping. Under this equivalence, comparing GR³ and LC-R1 reduces to:

**GR³:**

$$
\hat{R} = \begin{cases} 0 & R = 0 \\\ \dfrac{1}{1 + \alpha \dfrac{\ell}{\bar{\ell}}} & R = 1 \end{cases}
$$

**LC-R1:**

$$
\hat{R} = \begin{cases} 0 & R = 0 \\\ 1 + \lambda \left(1 - \dfrac{\ell}{\ell_{\max}}\right) & R = 1 \end{cases}
$$

|  | GR³ | LC-R1 |
| --- | --- | --- |
| **Length scale** | ℓ/ℓ̄ | ℓ/ℓ_max |
| **Normalization** | Group-relative | Global constant |
| **Penalty shape** | Nonlinear | Linear |
| **Reward range** | (0, 1] | [1, 1+λ] |

This reveals that the most salient difference may not be multiplicative vs. additive, but **group-relative normalization** (ℓ/ℓ̄) vs. a fixed global scale (ℓ/ℓ_max). Crucially, the main RLVR results (Table 3) where GR³ substantially outperforms LC-R1 fall exactly in the binary reward setting where the authors' own equivalence proof holds. Since the multiplicative vs. additive distinction vanishes here by construction, the performance gap may be attributed to other design differences (most plausibly the group-relative normalization).

Yet the paper provides no ablation to separate these factors. Figure 3's comparison bundles group-relative normalization into the multiplicative variant while using naive fixed-λ additive baselines, and comparisons against Efficiently and LC-R1 in Table 2 conflate multiple design axes simultaneously.

The authors' own analysis in Appendix B, showing that normalization anchor choice (mean vs. variance) critically affects stability, further suggests that the group-relative component may be equally or more important than the shaping paradigm.

To be clear, I do not want to challenge to the value of GR³ as a whole (group-relative normalization may well be a strong contribution in its own right). Rather, the concern is that the paper's narrative emphasizes multiplicative shaping as the primary innovation (Section 3.1, Propositions 3.1–3.2), yet the evidence does not cleanly separate its effect from that of group-relative normalization. A proper ablation would strengthen the paper by clarifying *which* component drives gains in *which* setting.

&nbsp;

Also, the paper seems to lack comparisons with recent length-compression baselines, such as "Sample More to Think Less: Group Filtered Policy Optimization for Concise Reasoning" (ICLR 2026), beyond settings where a simple length penalty is applied through the reward.

---

> ### Author Rebuttal · Authors · 2026-03-31
>
> Thank you for your recognition of the strengths of our work. We address your concerns as follows.
>
> **C1: The most salient difference may not be multiplicative vs. additive, but group-relative normalization vs. a fixed global scale ...  A proper ablation would strengthen the paper by clarifying which component drives gains in which setting.**
>
> A1: Thank you for recognizing the overall value of GR$^3$ and for your careful examination of the algorithm.
>
> We clarify that our emphasis on multiplicative shaping does not imply it alone is sufficient to achieve the full performance gains. The argument is fundamentally conceptual: rather than treating length as an external additive objective that competes with the original reward, or as a naive gated additive mechanism whose equivalence holds only in the binary RLVR setting, multiplicative shaping formulates length control as an adaptive modulation of the learning signal itself. It is important because it offers a more unified modeling paradigm for length control.
>
> Building on this foundation, we further incorporate group-relative normalization and advantage-aware calibration to achieve lossless optimization, as described in the paper, thereby better preserving model performance. To better disentangle the contribution of each component, we have added more comprehensive ablation studies:
>
> |Multiplicative Shaping|Our Group-Relative Term|Advantage-Aware Calibration|AIME 2024 Acc.|AIME 2024 Len.|AIME 2025 Acc.|AIME 2025 Len.|AMC 2023 Acc.|AMC 2023 Len.|MATH-500 Acc.|MATH-500 Len.|Avg. Acc.|Avg. Len.|
> |-|-|-|-|-|-|-|-|-|-|-|-|-|
> |✓|✓|✓|41.0|9089|29.2|8841|81.5|4513|86.6|2612|59.6|6264|
> |✓|✓|✗ ($\alpha$ too large)|32.1|5355|25.5|4667|69.3|1789|82.5|845|52.4|3164|
> |✓|✓|✗ ($\alpha$ too small)|40.9|12288|28.9|13115|80.0|7477|87.6|4046|59.3|9232|
> |✓|✗ (LC-R1 style)|✗|39.7|10391|28.2|10730|78.2|3254|81.6|1436|56.9|6453|
> |✓|✗ (Efficiently style)|✗|38.5|11041|29.3|11422|73.2|3944|81.2|1534|55.6|6985|
> |✗|✗ (LC-R1 style)|✗|25.1|5120|21.2|4097|68.5|2590|76.1|1488|47.7|3324|
> |✗|✗ (Efficiently style)|✗|26.6|5313|21.3|4503|67.5|2728|76.5|1506|48.0|3513|
>
> We highlight several details of the ablation. (i) The variant without multiplicative shaping but with the group-relative term is already shown in Figure 3 and omitted here. (ii) Advantage-aware calibration requires both shaping and group-relative components and is thus inapplicable when either is removed. (iii) For additive shaping, we use $\lambda=0.01$ based on empirical experience; a full sweep will be included in the revision. Due to the limited rebuttal timeline, all methods are trained for 500 steps and evaluated at that checkpoint.
>
> **C2: Lack comparisons with recent length-compression baselines, such as GFPO (ICLR 2026), beyond settings where a simple length penalty is applied through the reward.**
>
> A2: Thank you for raising this concern regarding the choice of baselines.
>
> We would like to clarify that the most critical baseline for length control is standard GRPO without any explicit length regularization, which is also the primary baseline adopted in GFPO (ICLR 2026). A key strength of our results lies in achieving comparable task performance to GRPO while significantly reducing response length, which we view as strong evidence of effectiveness.
> Regarding GFPO, we sincerely acknowledge its relevance and importance. However, as neither official code nor models are currently publicly available, direct reproduction within the rebuttal period is unfortunately infeasible. We fully agree that a direct comparison would be valuable and will include citation, thorough discussion, and experimental comparisons in the final version.
>
> In addition, we have reproduced DLER, a more recent method (Oct 2025), which provides a closer and practical comparison. The experimental results show that our method achieves better performance retention, further supporting the effectiveness of our approach.
>
> |Method|AIME 2024 Acc.|AIME 2024 Len.|AIME 2025 Acc.|AIME 2025 Len.|AMC 2023 Acc.|AMC 2023 Len.|MATH-500 Acc.|MATH-500 Len.|Avg. Acc.|Avg. Len.|
> |-|-|-|-|-|-|-|-|-|-|-|
> |GRPO|39.5|13305|30.5|12905|81.0|9177|87.6|5860|59.7|10312|
> |DLER|26.5|3442|22.0|3098|74.3|2221|84.5|1534|51.8|2573|
> |GR$^3$ (Ours)|41.0|9089|29.2|8841|81.5|4513|86.6|2612|59.6|6264|
>
> Please let us know if you have any additional questions.
>
> [1] Cheng Z, Chen D, et al. "Optimizing Length Compression in Large Reasoning Models." arXiv:2506.14755.
>
> [2] Arora D, Zanette A. "Training language models to reason efficiently." NeurIPS 2025.
>
> [3] Shrivastava V, Awadallah A, et al. "Sample more to think less: Group filtered policy optimization for concise reasoning." ICLR 2026.
>
> [4] Liu S Y, Dong X, et al. "DLER: Doing Length pEnalty Right-Incentivizing More Intelligence per Token via Reinforcement Learning." arXiv:2510.15110.

---

> > ### Author Rebuttal · Reviewer_PGQs · 2026-04-03
> >
> > Thank you for the thorough and detailed response. However, I remain not fully convinced regarding my primary concern, specifically that in binary reward settings typical of RLVR, the empirical gains may be largely driven by group-relative normalization rather than multiplicative shaping (or different hyperparameters), despite the paper's narrative emphasizing the latter as the central innovation. The additional ablation is helpful, but I am confused by the authors’ statement that “the variant without multiplicative shaping but with the group-relative term is already shown in Figure 3.” As I understand it, Figure 3 compares additive and multiplicative shaping, showing that additive shaping degrades reward while multiplicative shaping preserves it, rather than analyzing the effect of the group-relative term. Could the authors clarify this point?
> >
> > Regarding C2, my concern has been adequately addressed. Thank you.
> >
> > For these reasons, I would like to maintain my current rating.

---

> > > ### Author Response · Authors · 2026-04-06
> > >
> > > Thank you for your continued engagement with our work and for acknowledging our response regarding C2. We deeply appreciate your thoroughness, as your feedback helps us improve the clarity of our paper. We would like to address your remaining concern and clear up the confusion.
> > >
> > > 1. To clarify, our paper does not intend to claim that multiplicative shaping alone is sufficient to explain the full empirical gains. As stated in our rebuttal, our emphasis on multiplicative shaping is primarily conceptual. Rather than treating length control as an external additive objective that may easily compete with the original reward, or as a gated additive mechanism whose equivalence holds only in binary-reward settings, multiplicative shaping formulates it as an adaptive modulation of the learning signal itself. We believe this is an important modeling perspective because it provides a more unified and principled formulation of length control.
> > >
> > > 2. At the same time, we absolutely do not downplay the vital roles of the other two components. In our method, group-relative normalization and advantage-aware calibration are introduced on top of this multiplicative formulation to preserve the original optimization geometry and ensure lossless optimization. Accordingly, our core claim is that while multiplicative shaping serves as the formulation principle, the group-relative term and advantage-aware calibration are the crucial mechanisms driving the empirical performance gains. We will make this distinction more explicit in the final version.
> > >
> > > 3. Regarding our statement that “the variant without multiplicative shaping but with the group-relative term is already shown in Figure 3,” we sincerely apologize that our previous wording was insufficiently explicit.
> > > We made this statement to explain why we did not list the baseline [Multiplicative Shaping = ✗, Group-Relative Term = ✓] as a separate row in the new rebuttal ablation table. The reason is that this configuration corresponds to the curve in Figure 3 where task performance collapses. In that curve, the group-relative length term is retained, while the multiplicative formulation is replaced by an additive one. We will make this correspondence clearer in the final version.
> > >
> > > Thank you again for your thoughtful feedback. We hope this clarification resolves the remaining confusion and makes the role of each component more precise.

---

### Decision · Program_Chairs · 2026-04-30

**Decision:**

Accept (regular)

**Comment:**

This work introduces Group Relative Reward Rescaling (GR³), a framework that mitigates the length inflation problem in RL-trained LLMs by reframing length control as a multiplicative rescaling paradigm, combined with group-relative regularization and advantage-aware calibration.

The strengths of this research are listed as follows:
1. The multiplicative shaping can handle binary (RLVR) and continuous (RLHF) rewards, avoiding instability.
2. Advantage-aware calibration avoids temporary hyperparameter tuning.
3. In multiple domains, it reduced token usage while maintaining or improving standard performance.

The major concerns of this work are:
1. The core idea of multiplicative shaping has already been proposed in previous literature.
2. The exact contributions of multiplicative shaping and group-relative normalization need strict ablation experiments to distinguish.
3. Adopting open-source checkpoints as baselines might conflate methodological gains with different training settings.
4. Performance on out-of-distribution or more difficult datasets.

Added ablations, retrained baselines, and new OOD evaluations completely resolved the core concerns. Reviewer only suggests refining the camera-ready narrative to better distinguish its theoretical contributions from prior work.